## COMMUNICATION

# Effect of a sensing charge mutation on the deactivation of K$_V$7.2 channels

Baharak Mehrdel[1] and Carlos A. Villalba-Galea[1]

**Potassium-selective, voltage-gated channels of the K$_V$7 family are critical regulators of electrical excitability in many cell types. Removing the outermost putative sensing charge (R198) of the human K$_V$7.2 shifts its activation voltage dependence toward more negative potentials. This suggests that removing a charge "at the top" of the fourth (S4) segment of the voltage-sensing domain facilitates activation. Here, we hypothesized that restoring that charge would bring back the activation to its normal voltage range. We introduced the mutation R198H in K$_V$7.2 with the idea that titrating the introduced histidine with protons would reinstate the sensing charge. As predicted, the mutant's activation voltage dependence changed as a function of the external pH (pH$_{EXT}$) while modest changes in the activation voltage dependence were observed with the wild-type (WT) channel. On the other hand, the deactivation kinetics of the R198H mutant was remarkably sensitive to pH$_{EXT}$ changes, readily deactivating at pH$_{EXT}$ 6, while becoming slower to deactivate at pH$_{EXT}$ 8. In contrast, the K$_V$7.2 WT displayed modest changes in the deactivation kinetics as a function of pH$_{EXT}$. This suggested that the charge of residue 198 was critical for deactivation. However, in a surprising turn, the mutant R198Q—a non-titratable mutation—also displayed a high pH$_{EXT}$ sensitivity activity. We thus concluded that rather than the charge at position 198, the protonation status of the channel's extracellular face modulates the open channel stabilization and that the charge of residue 198 is required for the voltage sensor to effectively deactivate the channel, overcoming the stabilizing effect of high pH$_{EXT}$.**

## Introduction

Voltage-gated, potassium-selective (K$_V$) channels from the K$_V$7 family are common in the cardiovascular, gastrointestinal, and nervous systems. These proteins constitute the molecular entities responsible for M-currents that were first described as K$^+$-currents suppressed by the activation of Muscarinic receptors (Brown and Adams, 1980). Today, we know that K$_V$7 channels are regulated in many ways. For instance, K$_V$7 activity is tightly regulated by Gq protein-coupled receptors through the manipulation of phosphoinositides concentration, mainly PI(4,5)P$_2$ in the plasma membrane (Wang et al., 1998; Jentsch, 2000; Cooper, 2012). Furthermore, K$_V$7 channels are also subject to regulation by other lipids, including fatty acids (Taylor and Sanders, 2017; Larsson et al., 2020). Another outstanding property of K$_V$7 channels is that their activation is observed at membrane potentials as negative as –60 mV, while a typical K$_V$ channel activates at –40 mV or more positive potentials. This feature makes K$_V$7 channels critical determinants of the resting membrane potential in diverse cell types, dynamically regulating electrical excitability (Brown and Adams, 1980; Wang et al., 1998; Jentsch, 2000; Cooper, 2012; Linley et al., 2012; Soh et al., 2014; Mastrangelo, 2015; Miceli et al., 2015). Indeed, mutations that impair the normal functioning of K$_V$7 channels lead

to many types of disorders including benign familial neonatal seizures (Charlier et al., 1998; Singh et al., 1998; Dedek et al., 2001; Wuttke et al., 2008), early onset epileptic encephalopathy (Weckhuysen et al., 2012, 2013; Abidi et al., 2015; Mastrangelo, 2015; Miceli et al., 2015), peripheral nerve hyperexcitability (Dedek et al., 2001; Wuttke et al., 2007), and cardiac arrhythmias and long QT syndromes (Maljevic et al., 2010; Abbott, 2021; Sanguinetti and Seebohm, 2021).

Disruption of M-currents decreases the resting K$^+$-conductance in excitable cells of many types, facilitating the triggering of action potentials, thereby boosting excitability (Jentsch, 2000; Maljevic et al., 2010; Cooper, 2012; Abbott, 2021; Sanguinetti and Seebohm, 2021). Accordingly, poorly performing M-currents are seen as the culprit in the generation of out-of-control activity in a cell with K$_V$7 channels carrying disruptive mutations. One likely exception to this rule is the case of the K$_V$7.2 mutant R198Q. This mutation causes a negative shift in the activation voltage dependence of K$_V$7.2 channels (Millichap et al., 2017). Such shift results in an increased activity of the channel at the resting potential when expressed in *Xenopus laevis* oocytes (Millichap et al., 2017). Given the effect of the mutation R198Q on the channel's voltage dependence, the resting K$^+$ conductance should

[1]Department of Physiology and Pharmacology, Thomas J. Long School of Pharmacy, University of the Pacific, Stockton, CA, USA.

Correspondence to Carlos A. Villalba-Galea: cvillalbagalea@pacific.edu.



increase in neurons expressing this mutant channel, leading to hypoexcitability. However, R198Q is paradoxically deemed as a mutation linked to infantile spasms with hypsarrhythmia, a form of hyperexcitability-related disorder (Millichap et al., 2017). This led us to hypothesize that the shift of the voltage dependence might not be the reason for this mutation to cause seizures.

It has been reported that in a steady state at the resting potential, the open $K_V7.2$ channel becomes more resilient to be closed (Corbin-Leftwich et al., 2016; Villalba-Galea, 2020). Further, it has also been shown that both homomeric $K_V7.2$ and heteromeric $K_V7.2/K_V7.3$ channels become even more resilient to closing in the presence of Retigabine at low micromolar concentrations (Corbin-Leftwich et al., 2016). Furthermore, this "resilient-to-close" mode of activity is highly dependent on $PI(4,5)P_2$ as the deactivation is facilitated by the partial depletion of this signaling lipid (Villalba-Galea, 2020). These observations suggest that making the channels resilient-to-close (stabilizing the open conformations of the channels) is an essential feature in the physiology of these proteins.

It is well known that altering the sensing charges of a voltage-gated channel can have a strong impact on its voltage dependence and kinetics for activation. In contrast, much less is known about the effect of such alterations on channel deactivation. Thus, we proceeded to investigate whether "gain-of-function" mutations of residue R198 affect channel deactivation. Our study revealed that two non-charged R198 mutations of $K_V7.2$ gained another function as their deactivation kinetics became extremely sensitive to extracellular pH ($pH_{EXT}$). We observed that a small reduction of $pH_{EXT}$ (from 7.4 to 7.0) dramatically increased the deactivation rate. This observation was exciting because it has been shown that high neuronal activity can lead to a similar decrease in $pH_{EXT}$ during seizures (Raimondo et al., 2015; Sulis Sato et al., 2017). Considering this notion, we here propose that the mutation R198Q makes the open $K_V7.2$ channel less stable when the $pH_{EXT}$ becomes more acidic during periods of high neuronal activity, decreasing the ability of M-currents to curtail electrical excitability, thereby leading to seizures.

## Materials and methods

### Preparation of oocytes and RNA injections

RNA preparation, Xenopus oocyte isolation, preparation, and RNA injection were performed using published methods (Villalba-Galea et al., 2008, 2009; Corbin-Leftwich et al., 2016). Animal protocols were approved by the Institutional Animal Care and Use Committees at the University of the Pacific and conform to the requirements in the Guide for the Care and Use of Laboratory Animals from the National Academy of Sciences. Ovarian lobules were surgically harvested from frogs purchased from Xenopus 1. Oocytes were incubated at 16–17°C in a solution of (in mM) 100 NaCl, 1 KCl, 2 CaCl$_2$, 1 MgCl$_2$ or MgSO$_3$, 10 HEPES, 2 pyruvic acid, pH 7.5, and 20–50 mg/l of gentamycin. Results from different batches of oocytes were combined.

### Electrophysiology

Oocytes were injected with 2 ng of each in vitro-transcribed cRNA encoding for the human $K_V7.2$ and $K_V7.3$ channels.

Injected oocytes were incubated at 16–17°C for 2–4 days before recording. The incubation solution was titrated to pH 7.5 with NaOH and contained (in mM) 95 NaCl, 1 NaOH, 2 KCl, 5 HEPES, 1 MgCl$_2$, 1.8 CaCl$_2$, 1 MgCl$_2$, 2 pyruvic acid sodium salt, and 20–50 mg/l of gentamycin.

Potassium currents were recorded using the Xenopus oocyte Cut-Open Voltage-Clamp (COVC) technique with a CA-1 amplifier (Dagan Corporation). The external recording solution contained (in mM) 12 KOH, 88 N-methyl-D-glucamine, 85 methanesulfonic acid, 5 HEPES, 5 MOPS, 5 MES, 0.25 Mg(OH)$_2$, and 2 Ca(OH)$_2$. The external solution was titrated to pH 6.0, 7.0, 7.4, and 8.0 with methanesulfonic acid or NMDG, accordingly. The intracellular solution contained (in mM) 98 KOH, 2 KHPO$_4$, 88 methanesulfonic acid, 10 HEPES, 0.25 Mg(OH)$_2$, and 2 EGTA. The intracellular solutions were titrated to pH 7.4 with methanesulfonic acid. Borosilicate glass electrodes (resistance = 0.2–2.0 M$\Omega$) were filled with a solution containing (in mM) 1,000 KCl, 10 HEPES, and 10 EGTA, at pH 7.4 titrated with KOH.

As previously described, voltage control and current acquisition were performed using a USB-6251 multifunction acquisition board (National Instruments) controlled by an in-house program coded in LabVIEW (National Instruments; details available upon request). Current signals were filtered at 100 kHz, oversampled at 500 kHz–2 MHz, and stored at 5–25 kHz for offline analysis. Data were analyzed using custom Java-based software (details available upon request) and Origin 2019/Origin 2023b (OriginLab).

### Exponential fits and weighted average time constant

As described in previous studies (Wickenden et al., 2000; Labro et al., 2012; Villalba-Galea, 2014; Corbin-Leftwich et al., 2016), the following two-exponential function was fitted to the deactivating currents,

$$I_{DEACT}(t) = A_1 e^{-(t-t_0)/\tau_1} + A_2 e^{-(t-t_0)/\tau_2},$$

where, $A_1$ and $A_2$ are the current amplitude associated with each component, and $\tau_1$ and $\tau_2$ are the corresponding time constants. The parameter $t_0$ is the time at which deactivation starts. Fittings were done using Origin 2019 (OriginLab). When needed, the deactivation weighted average time constant ($\tau_{DEACT}$) was calculated as

$$\tau_{DEACT} = \frac{A_1\tau_1 + A_2\tau_2}{A_1 + A_2}.$$

T tests were calculated for statistical analysis of the time constants.

It is important to highlight that the two-exponential equation was not derived from a comprehensive kinetic model describing the activity of $K_V7$ channels. Instead, it was selected because it can tightly trace deactivating currents. Consequently, the parameters yielded from fitting the equation to such currents can only provide a temporal description of the deactivation process. Meaningful assignment of each individual parameter to any physical process underlying the activity of the channel under study is therefore extremely limited and even inadequate.

## Fermi–Dirac distribution, weighted $V_{1/2}$, and total apparent charge

The weighted $V_{1/2}$ was calculated from the values yielded by the fit of a double Fermi–Dirac distribution to the $I_{TAIL}$–$V_{ACT}$ plots. The double Fermi–Dirac distribution is defined as

$$I_{TAIL} = \frac{A_1}{1 + e^{-z_1 F(V_{ACT}-V_1)}/RT} + \frac{A_2}{1 + e^{-z_2 F(V_{ACT}-V_2)}/RT},$$

where $A_1$ and $A_2$ represent the amplitude of each of the components, $z_1$ and $z_2$ the apparent sensing charge associated with each component, and $V_1$ and $V_2$ are the half-maximum potential for each of the components. $F$, $R$, and $T$ are the Faraday constant, the universal gas constant, and the absolute temperature, respesectively. From here, the weighted $V_{1/2}$ is calculated as

$$weighted \ V_{1/2} = \frac{A_1 V_1 + A_2 V_2}{A_1 + A_2}.$$

The total apparent charge is given by

$$z_{apparent} = z_1 + z_2.$$

## Activation kinetic

To quantify the activation kinetics, we used an empirical equation that assumes that $n$ number of independent subunits are required to activate the conductance of the channels and that each subunit is activated in two steps. This is similar to what has been previously proposed for *Shaker* (Zagotta et al., 1994). We assumed the existence of a two-step process given that the activation displayed a fast and a slow component (e.g., Fig. 3 A). A two-step activation is represented by the sum of two exponential functions, and the requirement of >1 independent subunit for activation is represented by the power to $n$. The resulting equation is defined as

$$I_{ACT} = A\left[f_1\left(1 - e^{-\frac{t}{\tau_{FAST}}}\right) + (1 - f_1)\left(1 - e^{-\frac{t}{\tau_{SLOW}}}\right)\right]^n,$$

where A is the maximum amplitude for the currents, f1 represents the fractional contribution of the fast component, $t$ is time, and $\tau_{FAST}$ and $\tau_{SLOW}$ are the time constants of the fast and slow component, respectively.

## Molecular biology

The constructs human KCNQ2 and KCNQ3 in the expression vector pTLN, encoding $K_V7.2$ and $K_V7.3$ channels, were linearized with MluI and HpaI (New England Biolabs), respectively. The linearized $K_V7$-encoding cDNA was transcribed using an SP6 RNA polymerase kit (Ambion mMessage mMachine; Thermo Fisher Scientific). Mutations were introduced using the Q5 Site-Directed Mutagenesis Kit (New England Biolabs).

## Structural model

A cryo-EM-derived structural model of $K_V7.2$ bound to calmodulin (PDB ID 7CR3; Li et al., 2021) was incorporated in a POPC bilayer containing $PI(4,5)P_2$ using the online tool Charmm-Gui. Then, the model was minimized and equilibrated to 298.15° K using NAMD. A 10-ns simulation at 298.15° K was performed in the absence of any electrical field. No restrictions were imposed on the structure during the simulation. The

simulation was performed through an allocation grant from ACCESS.

## Statistical analysis

*T* tests were performed to determine significant differences between values by using the mean, SD, and number of observations to calculate P values. The confidence was set at 95%. *T* tests were performed using $pH_{EXT}$ 7.4 as a reference in figures plotting a parameter as a function of another parameter at different $pH_{EXT}$ (e.g., Fig. 3, D–I). *T* tests were performed using Origin 2023b (OriginLab).

## Online supplemental material

To account for potential electrostatic bias on the kinetic of activation, Fig. S1 shows the plots in Fig. 3, G–I, were replotted by changing the values of the X-axis at a rate of –17 mV/$pH_{EXT}$, using the $V_{ACT}$ values for the plot at $pH_{EXT}$ 7.4 as reference.

# Results

## Protonation of the R198H mutant biases voltage sensitivity for current activation

Mutating the arginine 198 to a glutamine (R198Q) causes a negative shift in the activation voltage dependence of $K_V7.2$ channels (Millichap et al., 2017; Edmond et al., 2022). To assess the effect of positive charges on residue 198 of the human $K_V7.2$, we proceeded to study the effect of the mutation R198H. Similar to what was observed with the residue R217 in Ci-VSP (Villalba-Galea et al., 2013), we were expecting to change the activation voltage dependence of the channel as a function of the protonation status of the introduced histidine if this manipulation effectively changed the charge of the residue. Following this approach, we recorded $K^+$ currents from oocytes expressing either wild-type (WT) human $K_V7.2$ and changed the extracellular pH ($pH_{EXT}$) between 6.0 and 8.0. Using the COVC technique, we applied voltage pulses ranging from –120 to +60 mV from a holding potential (H.P.) of –90 mV; deactivation was driven at –105 mV. In these recordings, we noticed that the activation of the $K_V7.2$ WT was almost impervious to changes in $pH_{EXT}$, showing small variations in kinetics (Fig. 1, A and B). However, the deactivation process was slightly more sensitive to $pH_{EXT}$ changes, slowing down at alkaline $pH_{EXT}$ (Fig. 1 B) with respect to acidic $pH_{EXT}$ (Fig. 1 A). For the mutant R198H, we applied a similar recording protocol. However, a 7-s prepulse to –120 mV preceded the test pulses, driving the closure of this mutant which displayed unambiguous activity at –90 mV at $pH_{EXT}$ above 7.0. When voltage steps above –90 mV were applied, the activation kinetics seemed almost unaltered by changes in $pH_{EXT}$. Surprisingly, however, the deactivation kinetics was remarkably dependent on the $pH_{EXT}$. Indeed, increasing $pH_{EXT}$ to 8.0 made the closing of the $K^+$ current (Fig. 1 D) unambiguously slower than that at $pH_{EXT}$ 6.0 (Fig. 1 C).

First, to further understand the effect of $pH_{EXT}$ on the activation of the mutant R198H, we plotted the normalized maximum amplitude of deactivating currents ($I_{TAIL}$) as a function of the amplitude of the activating membrane potential ($V_{ACT}$). The $I_{TAIL}$–$V_{ACT}$ plots showed a shift of about –40 mV at $pH_{EXT}$ 8.0

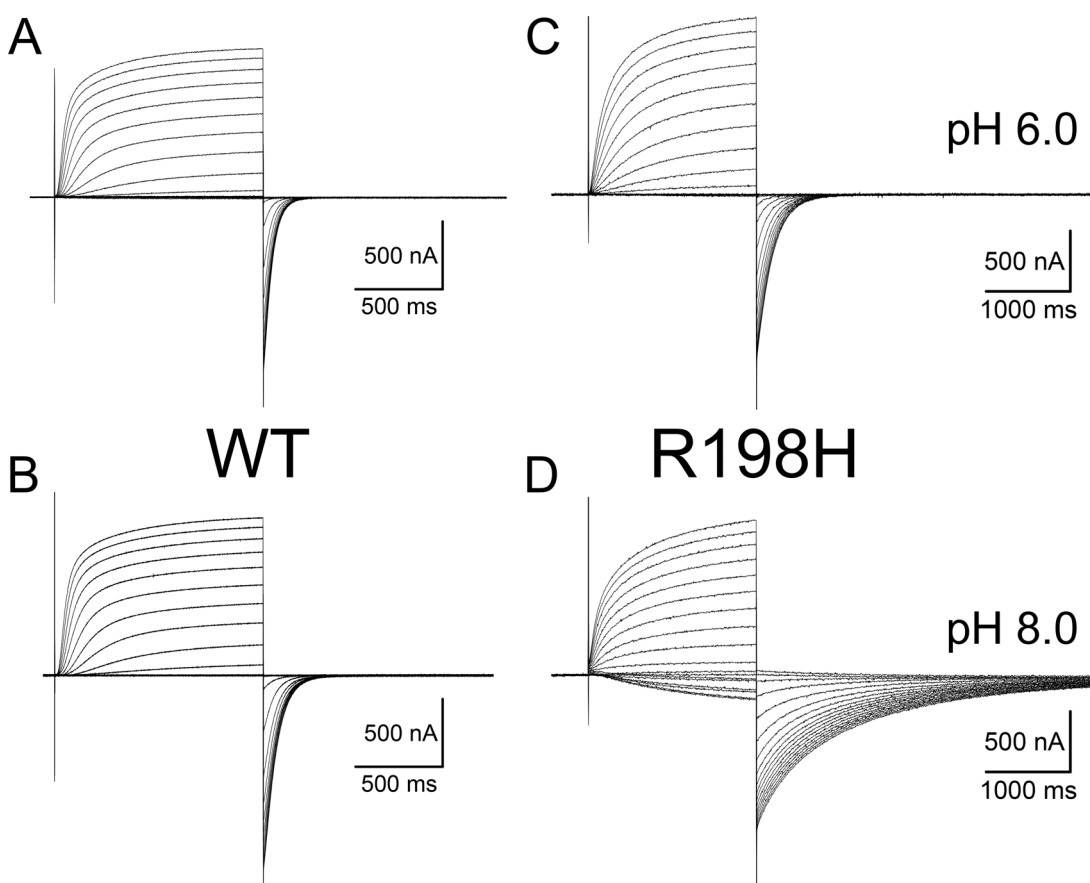

**Figure 1. Currents recording from oocytes expressing KV7.2 channels WT and mutant R198H. (A and B)** WT currents were activated with pulses ranging between −120 and +60 mV from a H.P. of −90 mV. **(C and D)** Equivalent recordings from oocytes expressing the R198H mutant with pulses ranging between −130 and +40 mV. A 5-s prepulse to −120 mV to these recordings as the voltage dependence of the mutant channel was shifted to more negative potentials with respect to WT. To evaluate the effect of external $pH_{EXT}$, we recorded currents at external pH ($pH_{EXT}$) ranging between 6.0 (A and C) and 8.0 (B and D). Remarkably, the rate of deactivation of R198H unambiguously decreased at pH 8.0 with respect to pH 6.0, while WT showed more modest changes.

(Fig. 2 B, purple squares) with respect to $pH_{EXT}$ 6.0 (Fig. 2 B, blue circle). In contrast, such changes in $pH_{EXT}$ had a modest effect on the WT channel activation (Fig. 2 A). This strongly suggested that the protonation of the histidine in position 198 was effectively altering voltage sensing during the activation of this channel. To quantify the effect of $pH_{EXT}$ on voltage dependence, we fitted a double Fermi–Dirac function to the $I_{TAIL}$–$V_{ACT}$ plots, not making any assumption of the underlying mechanism for voltage sensing. The Fermi–Dirac function is what is typically referred to as the "Boltzmann distribution." This latter name is a misnomer (Villalba-Galea and Chiem, 2020). Nonetheless, we calculated the average weighted half-maximum potential ($V_{1/2}$) for each $pH_{EXT}$ from the parameters yielded by the fit. We observed that the weighted $V_{1/2}$ values for the WT channels were modestly altered by changes in $pH_{EXT}$ (Fig. 2 C, black squares), displaying a 2 ± 1 mV change per unit of $pH_{EXT}$ (Fig. 2 C, black line: linear regression). In contrast, the weighted $V_{1/2}$ for the R198H mutant shifted to more negative potentials (Fig. 2 C, red circles), displaying a −17 ± 4 mV change per unit of $pH_{EXT}$ (Fig. 2 C, red line: linear regression). From the fits, we also observed that the apparent sensing charges (Fig. 2 D) and the fractional contributions of each component (Fig. 2 E) were barely altered

as a function of $pH_{EXT}$. Although these values for apparent charge cannot be used to calculate the actual sensing charge involved in voltage sensing (Bezanilla and Villalba-Galea, 2013), the fact that they are not altered strongly suggests that the intrinsic ability of the VSD to sense the electrical field was not modified by changes in $pH_{EXT}$. In other words, changes in the protonation status of the introduced histidine seemed to only produce an electrostatic effect, biasing the electrical field at the resting/deactivated state of the VSD.

**Effect of protonation on the activation kinetics**

To further understand the effect of the mutation R198H on the activation of KV7.2 channels, we proceeded to fit an $n$-powered two-exponential function to the activating currents as a function of time (Fig. 3 A). The two exponentials were used to fit the fast and slow components in the activation of these K+ currents (Fig. 3 A, teal and orange arrows, respectively). Furthermore, the function was elevated to the $n$-th power to be able to fit the lag phase of activation (Fig. 3 B, green arrow). It is generally expected that the parameter $n$ would tend to be close to 4 given the tetrameric nature of the KV channels (Zagotta et al., 1994)—this is indeed an expectation set since the 1950s,

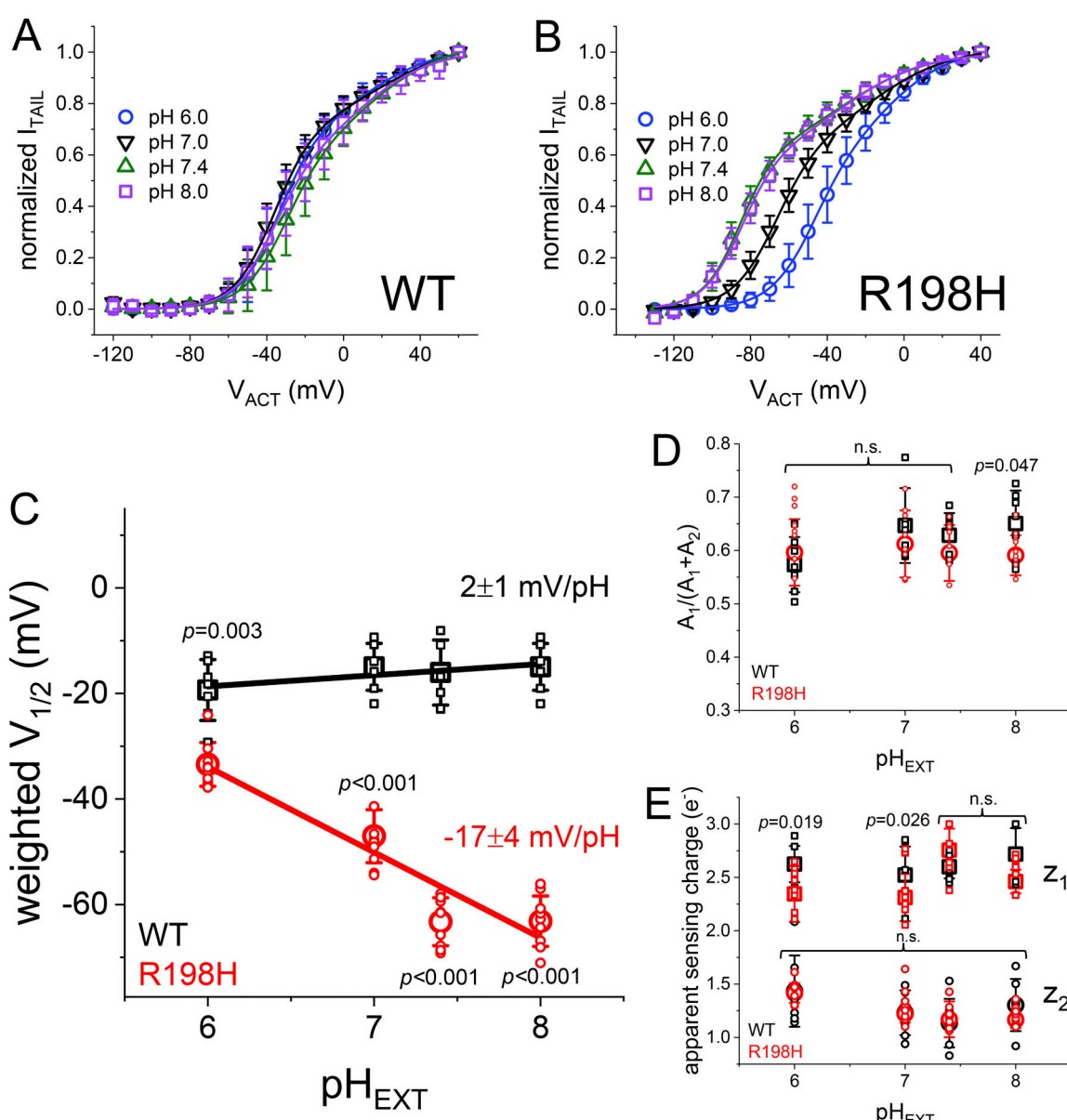

Figure 2. **Average normalized amplitude of deactivating K⁺-currents ("tail" currents, $I_{TAIL}$) plotted against the activating membrane potential ($V_{ACT}$).** **(A)** From oocytes expressing WT $K_V7.2$, the amplitude of $I_{TAIL}$ was normalized by its maximum peak current value in each individual recording. Then, the normalized $I_{TAIL}$ amplitudes were averaged and plotted as a function of $V_{ACT}$. The plots from recordings at $pH_{EXT}$ ranged from 6.0 to 8.0. **(B)** Equivalent $I_{TAIL}$–$V_{ACT}$ plots were generated from the recording of K⁺-current in oocytes expressing the mutant R198H. The $I_{TAIL}$–$V_{ACT}$ plot shifted about –40 mV at $pH_{EXT}$ 8.0 compared with $pH_{EXT}$ 6.0. **(C)** A double Fermi–Dirac equation was fitted to $I_{TAIL}$–$V_{ACT}$ plots from each experiment (see Materials and methods) and the weighted half-maximum of the $I_{TAIL}$–$V_{ACT}$ curves ($V_{1/2}$) and plotted as a function of $pH_{EXT}$. To quantify the $pH_{EXT}$-dependence of $V_{1/2}$, the $V_{1/2}$–$pH_{EXT}$ plots were fitted with a linear regression, yielding values of 2 ± 1 (mean ± SE, n = 7) and –17 ± 4 (n = 7), for WT and R198H, respectively. **(D and E)** Other two parameters derived from the Fermi–Dirac distribution were plotted against $pH_{EXT}$, namely the relative weight of the most negative component of the distribution (D) and the apparent charges associated with each component (E).

starting with the work of Hodgkin and Huxley in the squid axon (Hodgkin and Huxley, 1952). Yet, we observed that the values for n were around 3 for $K_V7.2$ WT and 2 for the mutant R198H. These observations suggested that the activation of less than four subunits is required for the ionic conduction to be elicited in these channels. This latter interpretation is consistent with the elegant work from the laboratory of Dr. Rene Barro-Soria (University of Miami), which suggests that not all VSD are to be activated to open the $K_V7.2$ channels conduction (Barro-Soria, R.,

personal communication). This is an intriguing idea that we did not further explore as it was beyond the scope of this work.

Focusing back on the kinetic of activation, the time constants for the fast and slow components for the WT channel, $\tau_{FAST}$, and $\tau_{SLOW}$, respectively, were almost impervious to changes in $pH_{EXT}$ (Fig. 3, D and E). Furthermore, the contribution of the fast component to the activation ($f_1$) was also resilient to changes in $pH_{EXT}$ (Fig. 3 F). This indicated that the activation process of the WT channel had a low $pH_{EXT}$ sensitivity. In contrast, the

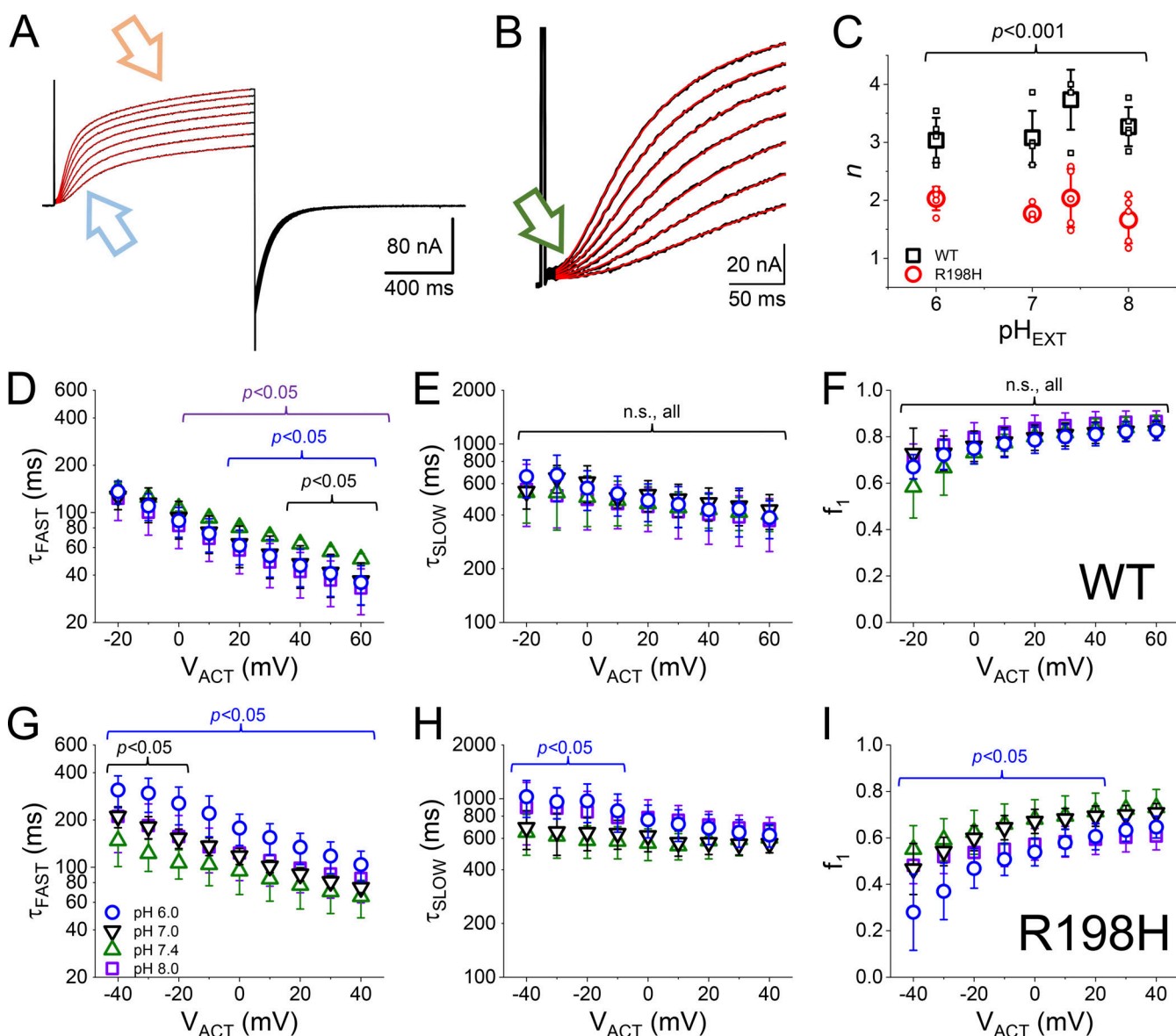

**Figure 3. Kinetics of K+-current activation. (A)** The activating current displayed two kinetic components, hereafter "fast" (teal arrow) and "slow" (orange arrow). **(B)** Also, the activation displayed a lag phase (green arrow). To accommodate these features, an $n$-powered double exponential function (red traces) was fitted to these activating K+-currents (black traces). **(C)** The index $n$ was different between WT and the mutant R198H, yet, it also seemed impervious to changes in $pH_{EXT}$. **(D and G)** Average time constant of the fast component ($\tau_{FAST}$) as a function of the activating potential ($V_{ACT}$) for both WT and R198H ($n = 5$, each). **(E and H)**. Average time constant as a function of the activating potential ($V_{ACT}$) for the slow component ($\tau_{SLOW}$) for both WT and R198H ($n = 5$, each). **(F and I)** Fraction of the fast component for the activation process for both WT and R198H ($n = 5$, each). Statistical test performed using $pH_{EXT}$ 7.4 as reference. T test confidences are color coded.

activation of the mutant R198H was unambiguously modulated by $pH_{EXT}$. The fast component seemed slower at $pH_{EXT}$ 6 (Fig. 3 G, blue circles), while the slow component seemed less sensitive to changes in $pH_{EXT}$ (Fig. 3 H). Furthermore, the fractional contribution of the fast component seemed to decrease at $pH_{EXT}$ 6.0 compared with more alkaline conditions (Fig. 3 I). Although the rates changed with $pH_{EXT}$, it is important to consider that the voltage dependence of activation also changes with $pH_{EXT}$ at an apparent rate of about –17 mV per unit of $pH_{EXT}$. This would make the voltage dependence of the activation rates at $pH_{EXT}$ 6.0 offset by 34 mV with respect to $pH_{EXT}$ 8.0. This indicated that the difference observed in the time

constants was likely due to the change in voltage dependence. Indeed, shifting the time constants versus potential curves for the mutant R198H (Fig. 3, G–I) made the difference between these curves to be more modest (Fig. S1). This led us to conclude that changes in electrostatic bias were the main effect of protonation on the kinetics of activation.

**Effect of $pH_{EXT}$ on deactivation**
To evaluate the effect of $pH_{EXT}$ on deactivation, we proceeded to study the deactivation kinetics as a function of activation using a previously established paradigm (Corbin-Leftwich et al., 2016; Villalba-Galea, 2020). Briefly, from a holding potential of

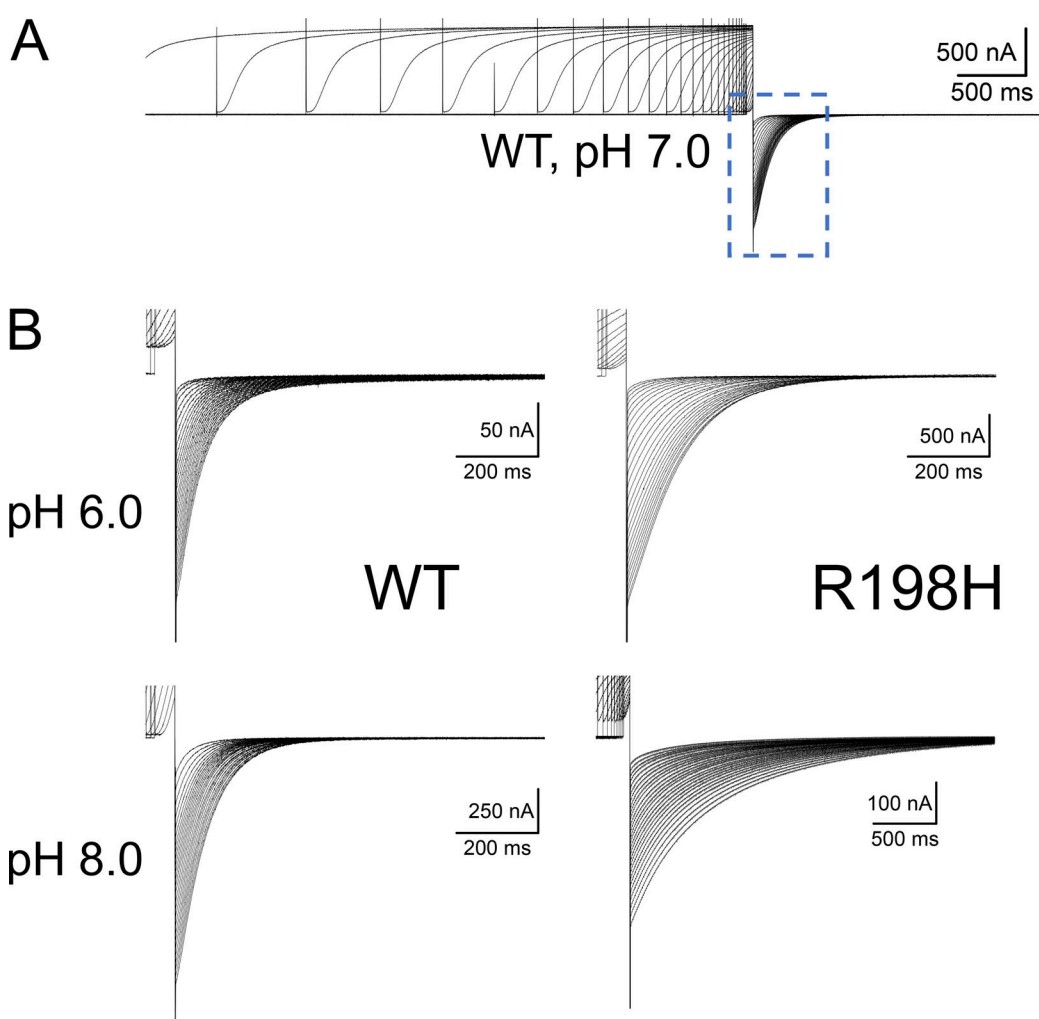

Figure 4.   **Deactivating of the K⁺-current from *Xenopus* oocytes expressing K$_V$7.2 channels WT and R198H at pHEXT 6.0 and 8.0. (A)** Example of K⁺-currents activated with a +40-mV pulse of exponentially increasing duration followed by deactivation at –105 mV. Deactivating phase inside dashed square. **(B)** Detailed deactivating K⁺-currents in oocytes expressing WT (left) and the mutant R198H (right) at pH$_{EXT}$ 6.0 (top) and 8.0 (bottom). Notice that the difference in the time scale for deactivation of the mutant R198H is at pH$_{EXT}$ 8.0.

–90 mV, we evoked K⁺ currents by applying +40-mV pulses of variable durations (Fig. 4 A), followed by a deactivating pulse to –105 mV (Fig. 4 A, dashed square). The deactivation kinetics was assessed during this second pulse to negative potentials. For the mutant R198H, this protocol was changed given that the currents were already activated at –90 mV. So, we set the H.P. to be close to the nominal reversal potential for K⁺ –50 mV (we had 12 mM K⁺ in the external solution and 100 mM K⁺ in the internal medium). The reason for setting the H.P. to this voltage was to avoid the accumulation of K⁺ in the extracellular side of the membrane. From this H.P., a 1.5-s ramp to –120 mV followed by a 7-s pulse to –120 mV was applied to close the channel. After that, a +40-mV pulse was applied to activate the channels. Then, deactivation was driven at –105 mV.

As reported before, the deactivation kinetics of K$_V$7.2 channels become slower as a function of the activating pulse durations (t$_{PULSE}$; Villalba-Galea, 2020). In the case of WT, the slowdown of the deactivation kinetics was slightly affected by changing pH$_{EXT}$ (Fig. 4 B, left panels). In contrast, deactivation of

the mutant R198H was remarkably slower at alkaline pH$_{EXT}$ with respect to acidic conditions and with respect to WT (Fig. 4 B, right)—notice time scale. To quantify the deactivation kinetics, a double exponential function was fitted to the monotonically decaying phase of the deactivating currents as previously described (Corbin-Leftwich et al., 2016; Villalba-Galea, 2020; Fig. 5 A, red traces). The weighted average time constants (τ$_{DEACT}$) were then plotted against t$_{PULSE}$ (Fig. 5). The τ$_{DEACT}$–t$_{PULSE}$ plot shows that deactivation became slower as the activating pulse was longer. As been previously reported (Corbin-Leftwich et al., 2016; Villalba-Galea, 2020), we observed that the slowdown of deactivation happens in two phases: an initial phase in which the deactivation time constant more than doubles t$_{PULSE}$ is <1 s and a late phase in which the deactivation changes in a more modest manner as a function of t$_{PULSE}$. For WT, we observed great variability in the second phase. Yet, changing pH$_{EXT}$ did not produce an unambiguous alteration of the τ$_{DEACT}$–t$_{PULSE}$ plot (Fig. 5 B). In contrast, the τ$_{DEACT}$–t$_{PULSE}$ relationship for the mutant R198H was extremely sensitive to

pH$_{EXT}$ (Fig. 5 C). In this latter case, increasing pH$_{EXT}$ caused a further decrease in $\tau_{DEACT}$, reaching a sevenfold increase in the deactivation time constant at pH$_{EXT}$ 8.0 with respect to pH$_{EXT}$ 6.0. Like in previous reports, this suggested that the slowdown of the deactivation was due to a transition into a more stable conductive/activated conformation of the channel and that such conformation was further stabilized in the R198H mutant at a higher pH$_{EXT}$.

**Absence of a positive charge in position 198 enhances K$_V$7.2 channel's pH$_{EXT}$ sensitivity**
To assess the need for a charge at position 198 to prevent enhancing K$_V$7.2 channels' pH$_{EXT}$ sensitivity, we engineered two additional mutations. The first one, R198K, introduced a residue that would be always charged within the range of pH$_{EXT}$ considered in this study. We observed that the weighted V$_{1/2}$ for the mutant was −26 ± 3 mV ($n$ = 7), which was negatively shifted with respect to WT with a weighted V$_{1/2}$ of −18 ± 5 mV ($n$ = 7). Although there was a shift in the voltage dependence of this mutant channel at pH$_{EXT}$ 7.4 (P = 0.0034), such property was almost impervious to changes in pH$_{EXT}$ like the WT (Fig. 6 A). This strongly suggested that the charge borne by the lysine at position 198 was able to prevent changes in the channel's activation by pH$_{EXT}$.

Coming full circle, the second mutation considered was R198Q. This is a clinically found mutation linked to spasms and seizures (Millichap et al., 2017). The mutation R198Q shifts the voltage dependence for activation of K$_V$7.2 channels toward negative potentials. As expected, we observed the weighted V$_{1/2}$ to be at −64 ± 7 mV and −64 ± 3 mV ($n$ = 7 for both) at pH$_{EXT}$ of 7.4 and 8.0, respectively. Yet, to our surprise, the weighted V$_{1/2}$ positively shifted to −52 ± 7 mV and −21 ± 6 mV ($n$ = 7 for both) at pH$_{EXT}$ 7.0 and 6.0, respectively (Fig. 6 B). This strongly suggested that, in the absence of a charged residue, the channel voltage-dependence becomes pH$_{EXT}$ sensitive independently of the protonation status of residue 198.

To further evaluate the effect of the mutations R198K and R198Q, we studied the deactivation kinetics of the K$^+$-currents. Like WT, the mutant R198K displayed fast deactivating currents at all pH$_{EXT}$ evaluated. We observed slight changes in the kinetic of deactivation, with the slowest kinetics being at pH$_{EXT}$ 8.0 ($n$ = 7; Fig. 6 C). In contrast, the mutant R198Q displayed a fivefold difference in the deactivation kinetics between pH$_{EXT}$ 6.0 and 8.0 (Fig. 6 D). These observations strongly suggested that a charged residue in position 198 allows the channels to overcome pH$_{EXT}$-related effects on the deactivation kinetics. We are yet to understand what residues participate in this enhancement in pH$_{EXT}$-sensitivity. Nonetheless, our results seem to indicate that having a positively charged residue at position 198 allows the channels to readily close.

**R198 mutants remain pH-sensitive when coexpressed with K$_V$7.3**
The effect of the mutation R198Q on the activation voltage dependence of K$_V$7.2 is mitigated by the coexpression of K$_V$7.3 (Millichap et al., 2017). Thus, we proceeded to evaluate how the presence of K$_V$7.3 affected the acquired pH$_{EXT}$ sensitivity. To do

so, we performed the same type of recordings described above from oocytes coexpressing the K$_V$7.3 channel subunit.

In the presence of K$_V$7.3, the voltage dependence of K$^+$ currents activation was less resilient to changes in pH$_{EXT}$ than the K$_V$7.2 channel expressed alone (Fig. 7, A and B). For the WT pair, the voltage dependence changed by a rate of −5 ± 1 mV/pH unit ($n$ = 5–7; Fig. 7 G), becoming slightly more positive as the pH$_{EXT}$ becomes acidic (Fig. 7, left). In contrast, the voltage dependence of activation of the mutants R198H (Fig. 7, C and D) and R198Q (Fig. 7, E and F) retained strong pH sensitivity. The mutant R198H shifted its voltage dependence −17 ± 2 mV/pH unit ($n$ = 8–10; Fig. 7 G) like the mutant R198H expressed alone. Similarly, the mutant R198Q displayed a shift rate of −14 ± 1 mV/pH unit ($n$ = 6–12; Fig. 7 G). For the three cases, changing pH$_{EXT}$ barely altered the total apparent charge associated with activation (Fig. 7 H). These observations suggested that the modulation of the activation voltage dependence might be an intrinsic property of each subunit as the gained-by-mutation pH$_{EXT}$ sensitivity remained in their activity when coexpressing the K$_V$7.3 subunit.

Regarding deactivation, the coexpression of K$_V$7.3 did not prevent the pattern observed above. K$^+$ currents from oocytes expressing K$_V$7.2-WT and K$_V$7.3 showed minor changes in their deactivation kinetics as a function of pH$_{EXT}$. In a remarkable contrast, the deactivation kinetics of each of the mutants K$_V$7.2-R198H and K$_V$7.2-R198Q coexpressed with K$_V$7.3 were sensitive to pH$_{EXT}$. In both cases, increasing pH$_{EXT}$ strongly decreased the rate of deactivation (Fig. 7, A, C, and E). These observations indicated that the coexpression of the K$_V$7.3 subunit could not override the effect of neutralizing the charge at position 198.

**The countercharge E130 is not responsible for the pH$_{EXT}$ sensitivity**
Observing that mutant R198Q was highly sensitivity to changes in pH$_{EXT}$ suggested that the alterations of the voltage dependence for activation and deactivation kinetics did not emerge primarily from the altering of the protonation/electrostatic status of the residue at position 198. This led us to propose that other residues are responsible for the pH$_{EXT}$ sensitivity acquired by the mutant channels. Among other potential candidates, we focused on one of the negative charges commonly found in the S2 segment of the VSD of channels, as suggested by reviewers of this work. To explore this idea, we looked at one of the recent cryo-EM-derived structural models reported for K$_V$7.2 (Li et al., 2021). This model was incorporated in a membrane using Charmm-Gui (https://www.charmm-gui.org/), equilibrated and relaxed in a 10-ns simulation using NAMD (University of Illinois at Urbana-Champaign). We notice that glutamate at position 130 (E130) was close to arginine 198 of the model (Fig. 8). This suggested that titration of this acidic residue could be responsible for the acquired pH$_{EXT}$-sensitivity of the mutants R198H and R198Q. We reasoned that the positive charge carried by R198 interacts with the negative charge of E130 in the down/deactivated state of the VSD. If the charge in position 198 is not present, then the putative interaction between E130 and R198 will not exist, favoring the activated state of the VSD. If this is the case, then removing the negative charge borne by residue 130 should have a similar effect than removing the charge at residue

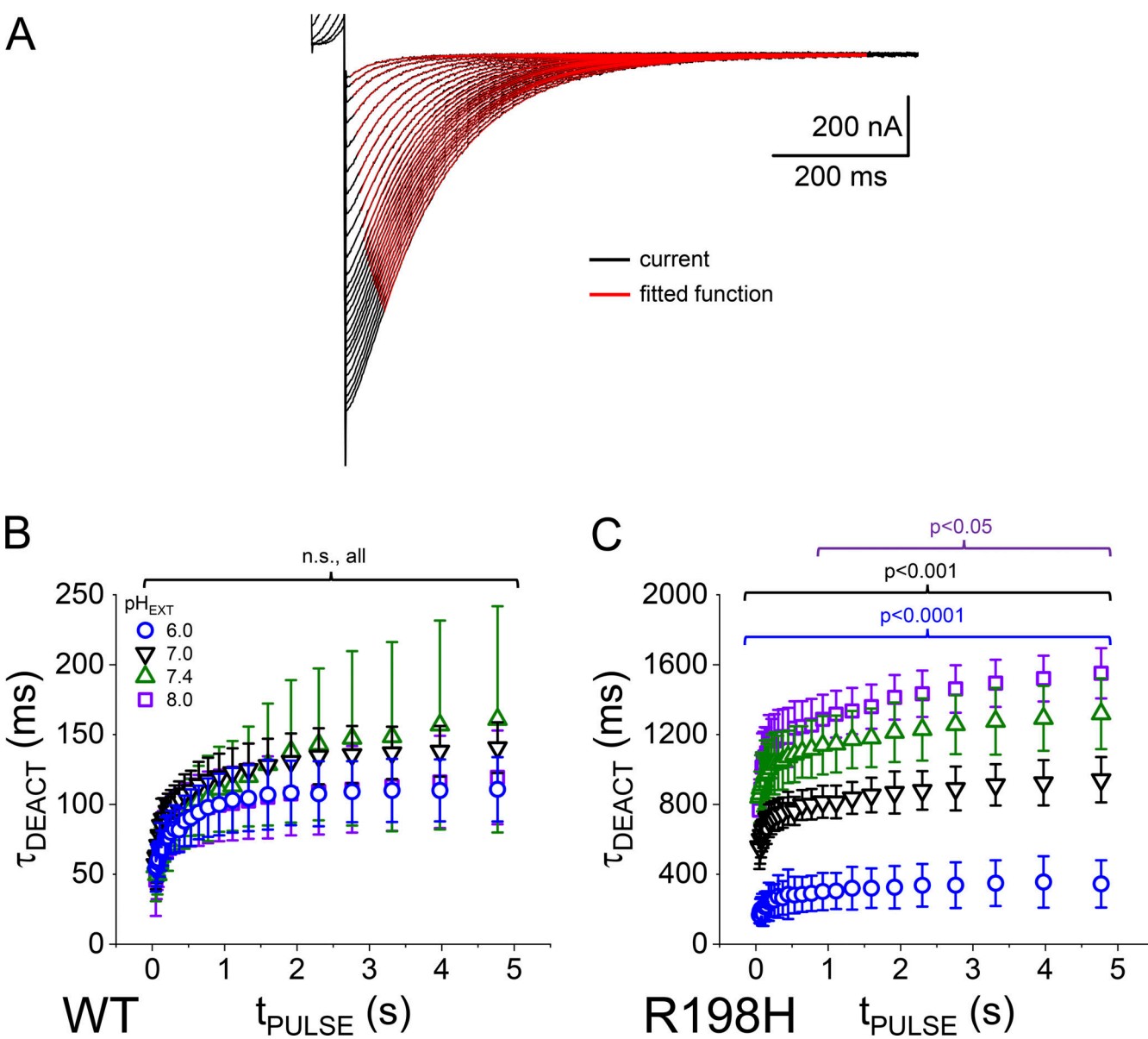

Figure 5. **Analysis of the deactivation time constant. (A)** A two-exponential function was fitted to the deactivating K⁺-currents. The monotonically decaying phase of the deactivating currents was fitted with a double exponential function (red traces); there is a small yet unambiguous lag phase at the beginning of the deactivation currents. **(B and C)** The average deactivation time constants ($\tau_{DEACT}$) were plotted as function of the duration of the activating pulse ($t_{PULSE}$) for oocytes expressing WT $K_V7.2$ (B, $n = 9$) and the mutant R198 (C, $n = 16$ for $pH_{EXT}$ 6.0, $n = 11$ for $pH_{EXT}$ 7.0, $n = 8$ for $pH_{EXT}$ 7.4, and $n = 6$ for $pH_{EXT}$ 8.0). Statistical test was performed using the $pH_{EXT}$ 7.4 as reference. $T$ test confidences are color-coded.

198. To assess this hypothesis, we used the same approach described above and recorded K⁺ currents from oocytes expressing a mutated $K_V7.2$ in which the residue E130 was replaced with a histidine. The idea was that if the charge of residue 130 is important, then titration by changing $pH_{EXT}$ will allow modulation of the activity of the channel by protonation. Yet, we observed that the mutant E130H was not readily expressed in our oocytes (Fig. 9 A, top). This strongly suggested that the mutation E130H was not tolerated. This was surprising because channels bearing mutations like E130R and E130C can produce currents (Soldovieri et al., 2019). Nonetheless, we did not explore this issue any further. Instead, we proceeded to coexpress the mutant E130H with $K_V7.3$. It has been well-established that the expression of $K_V7.3$

alone in *Xenopus* oocytes results in small K⁺ currents (Wang et al., 1998; Etxeberria et al., 2004). Thus, if the co-expression with the mutant E130H and $K_V7.3$ produced currents, that would mean that the presence of $K_V7.3$ boosted the expression of the mutant subunit. Luckly, we observed a robust expression of K⁺ currents when expressing these constructs together, suggesting that the presence of the sister subunit compensated for the structural deficiency introduced by the mutation. Beyond this, we observed that the mutation did not increase the $pH_{EXT}$ sensitivity of the channel (Fig. 9 A, middle and bottom). In fact, the average normalized $I_{TAIL}$–$V_{ACT}$ plots at different $pH_{EXT}$ overlapped (Fig. 9 B). Furthermore, fitting a double Fermi–Dirac distribution to individual $I_{TAIL}$–$V_{ACT}$ shows that the average weighted $V_{1/2}$ only

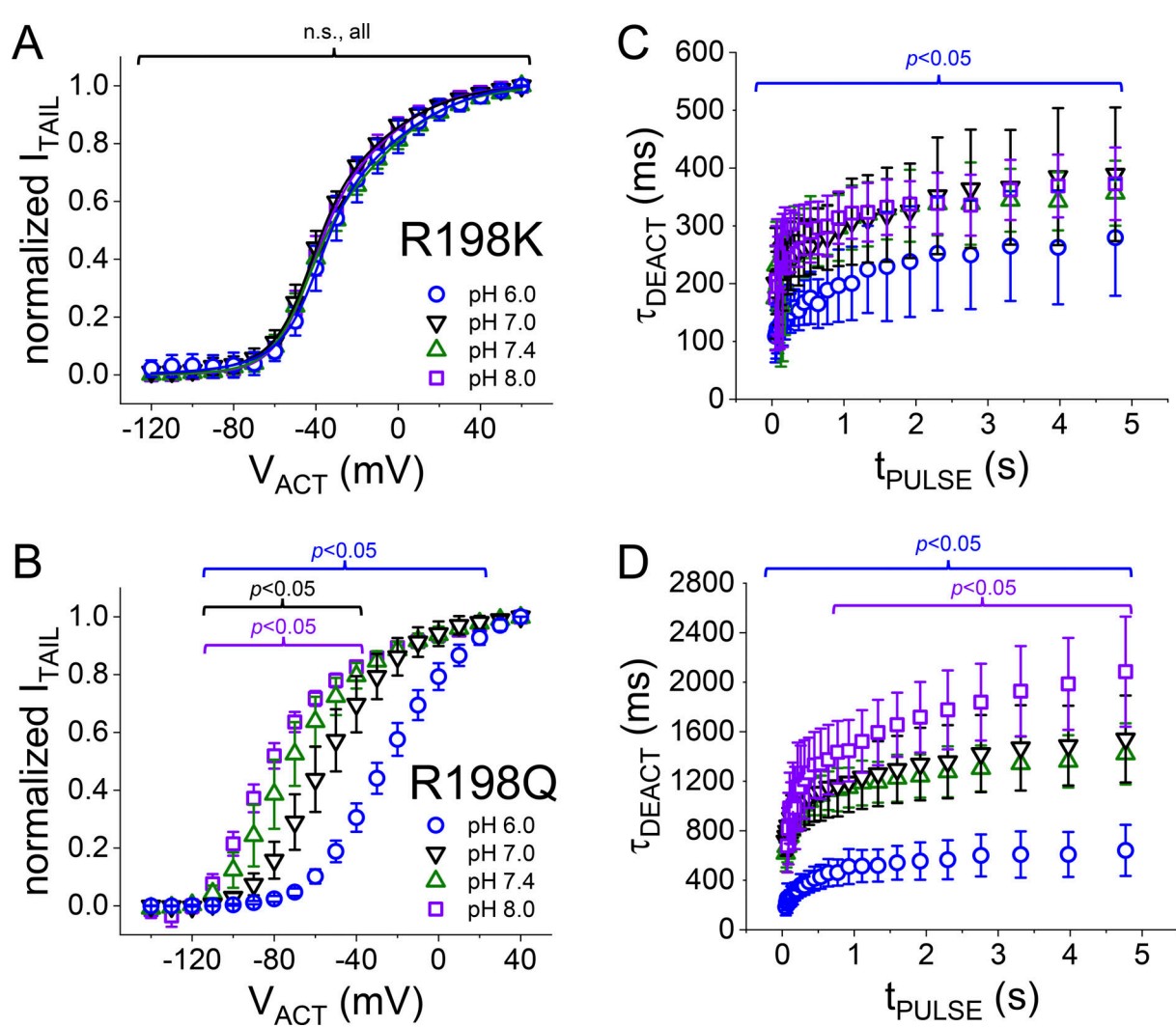

Figure 6. **Activation voltage dependence and deactivation kinetics of the mutants R198K and R198Q. (A and B)** $I_{TAIL}$–$V_{ACT}$ plots generated from $K^+$-currents recorded from oocytes expressing the mutants R198K (A) and R198Q (B) at $pH_{EXT}$ ranging from 6.0 to 8.0. **(C and D)**. Deactivation average time constant ($\tau_{DEACT}$) for the mutants R198K ($n$ = 8) (C) and R198Q (D) also at $pH_{EXT}$ 6.0 through 8.0. Statistical test performed using the WT as reference. $T$ test confidence intervals are color-coded; $n$ = 8 for $pH_{EXT}$ 6.0, $n$ = 11 for $pH_{EXT}$ 7.0, $n$ = 12 for $pH_{EXT}$ 7.4, and $n$ = 7 for $pH_{EXT}$ 8.0.

changed at a rate of –4 mV per unit of $pH_{EXT}$ ($n$ = 9–12; Fig. 9 C). This was like what we observed with the pair $K_V7.2$/$K_V7.3$. Finally, changes in $pH_{EXT}$ barely altered the apparent total charge associated with the voltage dependence of activation (Fig. 9 D). These combined observations indicated that E130 was not involved in changing the voltage sensitivity of the channel as a function of $pH_{EXT}$ and that it only shifted the overall voltage dependence for activation by about +20 mV.

In addition to the lack of effect of change in $pH_{EXT}$ on the activation of the mutant E130H, we also observed the deactivation was also not $pH_{EXT}$ sensitive (Fig. 10 A). Fitting a double exponential function to the deactivating currents shows that changes in $pH_{EXT}$ did not alter the $\tau_{DEACT}$–$t_{PULSE}$ relationships (Fig. 10 C).

## Discussion

Here, we have shown that two charge-neutralizing mutations of arginine 198 asymmetrically affect the activation and

deactivation of the human $K_V7.2$ channel. On one hand, removal of the charge at position 198 by mutation caused a shift in voltage dependence for activation, which is consistent with a previous report (Millichap et al., 2017). This shift also caused a concomitant yet small alteration of the activation kinetics. On the other hand, the charge-neutralizing mutations produced remarkable changes in the deactivation kinetics and conferred a higher $pH_{EXT}$ sensitivity to the mutant channel's activity. This asymmetrical effect on $pH_{EXT}$ sensitivity highlights the notion that, at least for $K_V7.2$ and $K_V7.2$/$K_V7.3$ channels, the activation and deactivation processes follow distinct pathways (Corbin-Leftwich et al., 2016; Villalba-Galea, 2020).

The residue R198 is in the fourth (S4) segment of the VSD, where it is the outermost arginine of the segment (Li et al., 2021). In that position, R198 could function as one of the sensing charges of the channel's VSD. Adding and removing charges in the S4 segment can cause changes in voltage dependence, kinetics, and the appearance of additional conductances

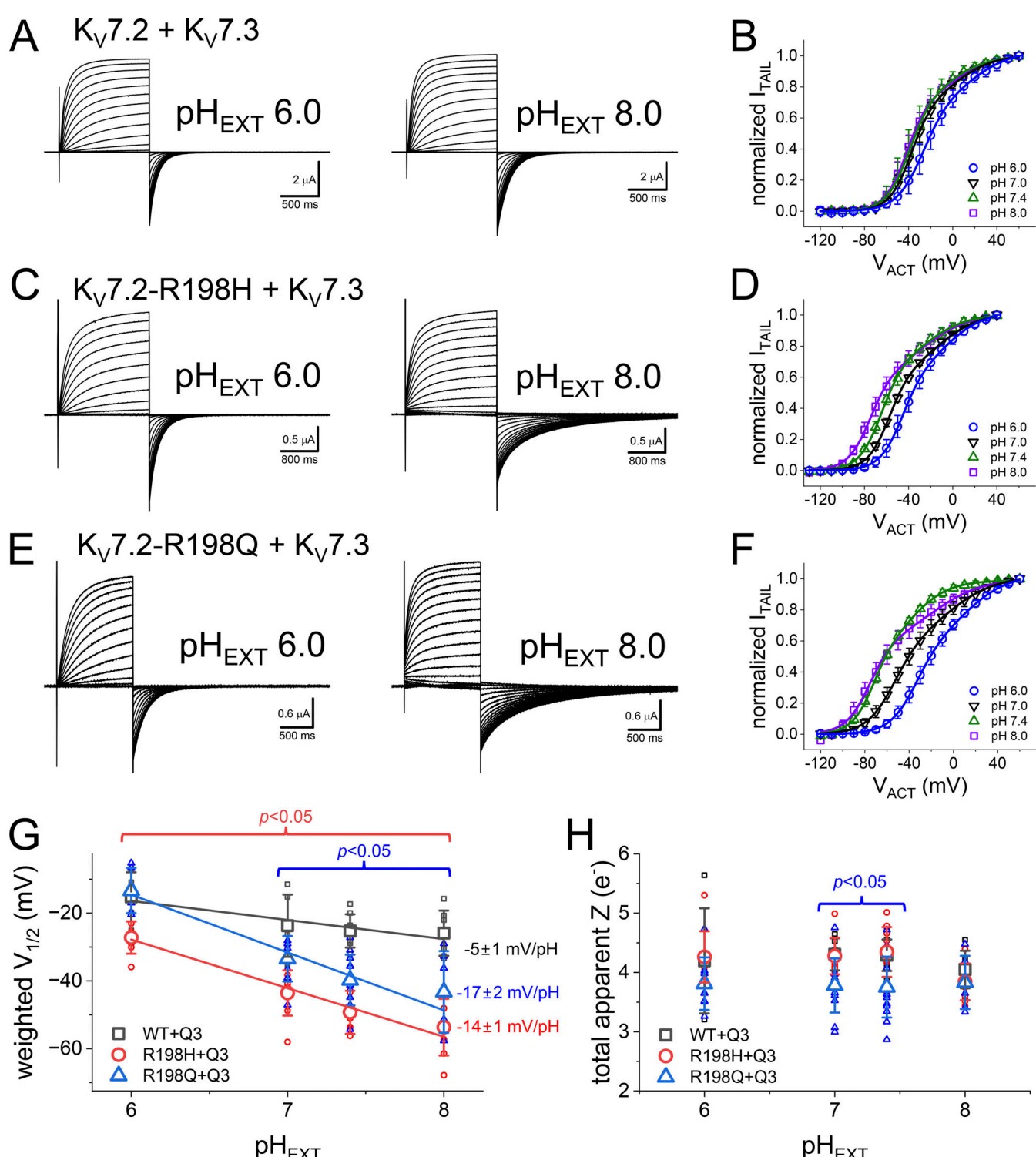

Figure 7. **Effect of the coexpression of $K_V7.3$ on the deactivation kinetics at different $pH_{EXT}$. (A)** $K^+$ current recordings from oocytes expressing the WT $K_V7.2$ channel coexpressed with $K_V7.3$. The recordings were performed at $pH_{EXT}$ 6.0 (left) and 8.0 (right). The holding potential was set to −90 mV, and the amplitude of the activating test pulses ($V_{ACT}$) ranged from −120 to +60 mV. Deactivation was driven at −105 mV. **(B)** Average $I_{ACT}$–$V_{ACT}$ plots at $pH_{EXT}$ ranging from 6.0 to 8.0. **(C)** $K^+$-currents from oocytes coexpressing the mutant R198H with $K_V7.3$. In this case, the holding potential was set at −50 mV, which was close to the nominal Nernst potential of $K^+$ in the experimental conditions. The test pulses ranged from −140 to +40 mV. Before the test pulses, the membrane potential was set to −120 mV. **(D)** Average $I_{TAIL}$–$V_{ACT}$ for the mutant R198H coexpressed with $K_V7.3$. **(E)** $K^+$ current recording from oocytes coexpressing the R198Q mutant along with $K_V7.3$ using the same protocol in C. **(F)** Average $I_{TAIL}$–$V_{ACT}$ plots for the mutant R198Q coexpressed with $K_V7.3$. Solid lines in B, D, and F are the Fermi–Dirac function fitted to each average $I_{TAIL}$–$V_{ACT}$ plot. **(G)** Average weighted $V_{1/2}$ for each of the WT and the mutants R198H and R198Q each coexpressed with the $K_V7.3$. The weighted $V_{1/2}$ was calculated from the parameters yielded from fitting a double Fermi–Dirac function to the $I_{TAIL}$–$V_{ACT}$ from individual experiments. The solid lines correspond to a linear regression of the $V_{1/2}$ as a function of $pH_{EXT}$. We used a linear function to merely characterize the changes in $V_{1/2}$ not assuming any type of mechanism. **(H)** Average total apparent charge associated with gating. Statistical test performed using the WT as reference. $T$ test confidence intervals are color coded. For WT: $n$ = 5–9; for R198H: $n$ = 8–12; for R198Q: $n$ = 6–14.

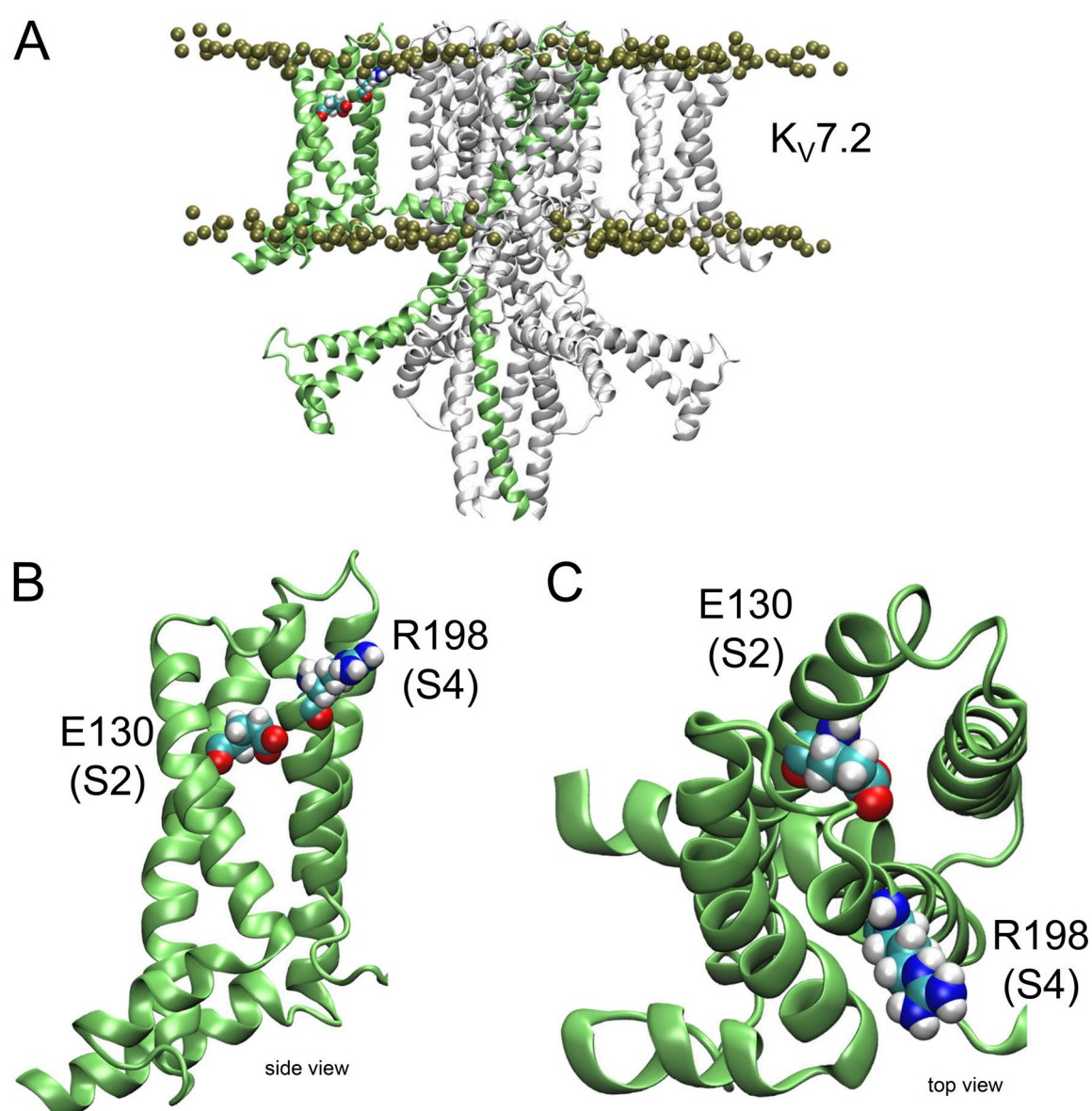

**Figure 8.** **Structural model of K_V7.2 derived from the cryo-EM structure 7CR3. (A)** Full view of the tetrameric model embedded in a POPC bilayer (phosphate group depicted as tan spheres). One of the subunits is highlighted in green. **(B and C)** Detailed side (B) and top (C) views of VSD's backbone from the highlighted subunit in A. Residues E130 and R198 are depicted as van der Waals spheres.

(i.e., "omega"/"gating pore" currents), among other loss-of-function and gain-of-function alterations (Ahern and Horn, 2004; Starace and Bezanilla, 2004; Sokolov et al., 2007; Catterall, 2010; Zhao and Blunck, 2016; Gamal El-Din et al., 2021). Varying the residue composition of the voltage sensor of a channel, particularly S4 residues, has resulted in changes in the activity of channels that cannot be explained only in terms of basic electrostatic (Starace et al., 1997; Starace and Bezanilla, 2001; Ahern and Horn, 2004; Sokolov et al., 2007; Yang et al., 2007, 2011; Gagnon and Bezanilla, 2009; González-Pérez et al., 2010). These types of alterations in the properties of the channels are due to remodeling of the sensor so that changes in local hydrophobicity, hydration, and steric hindrance play a critical role along with changes in the electrostatic of each mutated residue. Here, the introduction of a histidine at the top

of the S4 segment (mutation R198H) granted the channel with high $pH_{EXT}$ sensitivity. Decreasing $pH_{EXT}$ shifted the voltage dependence of activation of the charge-neutralized mutants to a more positive potential, while increasing $pH_{EXT}$ had the opposite effect. To our surprise, it was not the ability to titrate the histidine with protons but the lack of the charge itself that conferred high $pH_{EXT}$ sensitivity as the non-titratable mutation R198Q displayed a pH-dependent behavior like the mutant R198H.

A recent study showed that the spasm/epilepsy-related mutation R198Q causes a shift in voltage dependence to more negative potential, while decreasing the slope of the current amplitude-vs.-membrane potential curve, supporting the idea that R198Q is likely the first sensing charge. However, careful fitting of the current-vs.-voltage relationship (i.e., $I_{TAIL}–V_{ACT}$ plots in Fig. 2) indicated that the "steepness" of these curves did

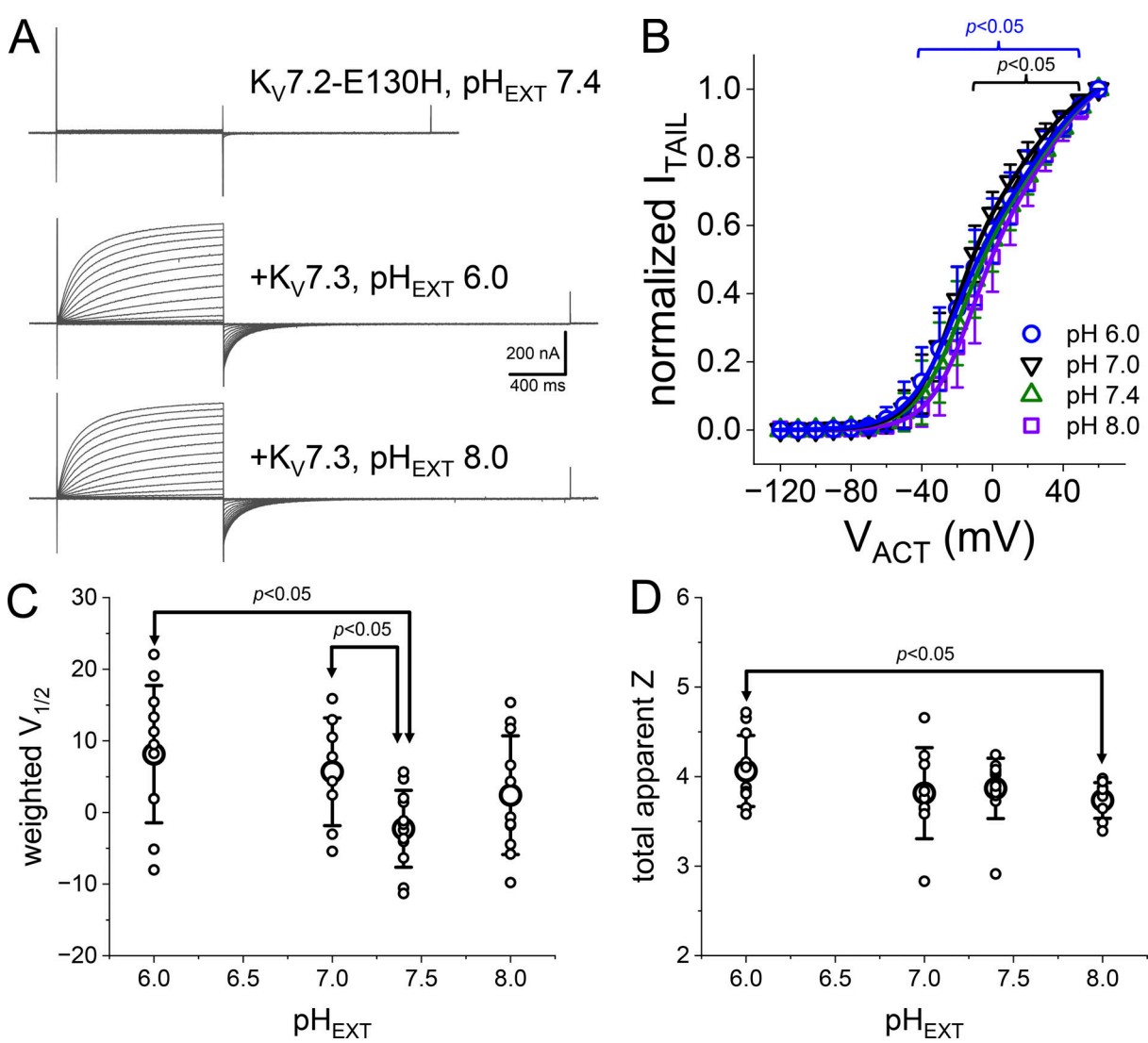

Figure 9. **K+ currents from oocytes expressing the mutant E130H. (A)** Examples of currents recorded from oocytes expressing the mutant E130H alone (top) and with $K_V7.3$ (middle and bottom). **(B)** Average $I_{TAIL}$–$V_{ACT}$ plots for the mutant E130H coexpressed with $K_V7.3$ ($n$ = 9–12). Currents recorded at $pH_{EXT}$ 6.0, 7.0, 7.4, and 8.0. **(C and D)** Average weighted $V_{1/2}$ and average total apparent charge calculated from individual normalized $I_{TAIL}$–$V_{ACT}$ plots.

not change with $pH_{EXT}$, strongly suggesting that R198 does not play a role as a sensing charge. Although the "slope" is a poor indicator of the number of charges involved in conferring voltage dependence to channels (Bezanilla and Villalba-Galea, 2013), the fact that changing the $pH_{EXT}$ did not alter the steepness of the $I_{TAIL}$–$V_{ACT}$ plots supports the idea that R198 might not be playing a role as a first sensing charge.

Another argument against, yet not proving, the idea that R198 is the first sensing charge in the VSD of $K_V7.2$ is the absence of inwardly rectifying currents at negative potentials. If R198 was the first sensing charge, the mutation R198H could open a conduction pathway, producing a down-state current through the VSD like in the case of sodium-selective voltage-gated ($Na_V$) channels (Sokolov et al., 2007), the $K_V$ channel *Shaker* (Starace et al., 1997; Starace and Bezanilla, 2001), the voltage-sensitive phosphatase (Ci-VSP; Villalba-Galea et al., 2013), and $K_V7.3$ channels (Gamal El-Din et al., 2021). Again, the absence of such currents does not prove that R198 is not the first sensing charge

of the VSD's S4 segment. Nonetheless, we propose that R198 plays a critical role in the modulation of channel activity while not directly participating in voltage sensing.

## Asymmetric effect of charge-neutralizing mutations of R198 on activation and deactivation

A remarkable feature of both R198H and R198Q is that the effect of $pH_{EXT}$ on their deactivation kinetics seems to be "out of proportion" with respect to that on the activation kinetics. In the case of the mutant R198H, changes in $pH_{EXT}$ modestly affected the activation kinetics compared with the changes observed in deactivation. For activation, the effect of changes in the $pH_{EXT}$ seems to cause electrostatic effects, at least for the mutant R198H. In fact, replotting the values of $\tau_{FAST}$, $\tau_{SLOW}$, and $f_1$ for this mutant (Fig. 3, G–I) and adding a bias of –17 mV/$pH_{EXT}$ units made those relationships almost overlap (Fig. S1). This suggested that the protonation of R198H mostly produced an electrostatic effect on voltage sensing. However, the

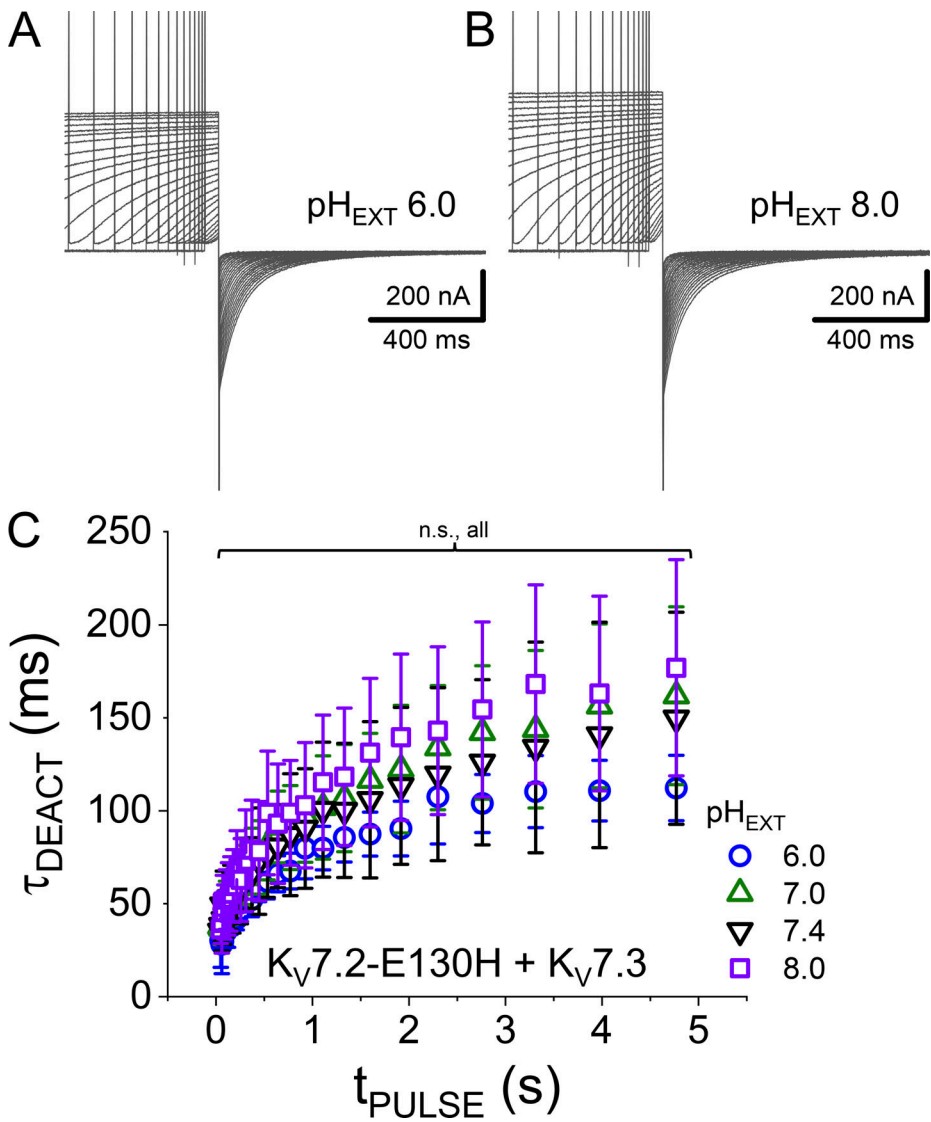

**Figure 10.** **Deactivation of the mutant E130H coexpressed with K$_V$7.3. (A and B)** Examples of recordings of K$^+$ current deactivation at pH$_{EXT}$ 6.0 (A) and 8.0 (B) following activation pulse of variable duration to +40 mV. Deactivation was driven by a pulse to −105 mV. **(C)** A double exponential function was fitted to the deactivating currents (not shown). The yielded parameters were used to generate average τ$_{DEACT}$–t$_{PULSE}$ plots at different pH$_{EXT}$. (pH$_{EXT}$ 6.0: $n = 5$; 7.0: $n = 3$; 7.4: $n = 6$; 8.0: $n = 5$).

"protonation hypothesis" does not explain what happened with the mutant R198Q. Thus, we can only conclude at this point that the presence of a positively charged residue in position 198 allows the channels to overcome the stabilization of the open channel when increasing pH$_{EXT}$.

Regarding deactivation, the kinetics of this process for the charge-neutralized mutants displayed larger differences of four and sevenfold as a function of pH$_{EXT}$. This indicated that the process of activation and deactivation follow distinct pathways so that they are not each other's reverse processes. This is consistent with previous observations, in which the effects of K$_V$7 channel modulator retigabine and the regulation by the phosphoinositide PI(4,5)P$_2$ seem to differentially affect activation and deactivation (Villalba-Galea, 2020).

It is not clear what the mechanism underlying the asymmetrical effect of pH$_{EXT}$ on the channel's kinetics is. Here, we

propose that at high pH$_{EXT}$ the open conformations of the channel become resilient to closing. However, the presence of a charge at position 198 allows for an effective voltage-driven deactivation of the channel. Alternatively, a charged residue in position 198 could be part of a network of salt bridges, making the channel less susceptible to protonation. The absence of this charge disrupts this hypothetical network, causing the rearrangement of the VSD and so changing voltage dependence for activation. Although distinguishing between these and other mechanisms escapes the scope of this study, we proceeded to alter one obvious candidate residue that could be a member of this putative network. That residue E130 is one of the negative charges found in S2 of the VSD, it could be interacting with R198 in the deactivated conformation of the VSD since E130 faces the extracellular crevice of the domain (Fig. 8). We observed that mutant E130H was not readily expressed in oocytes. This

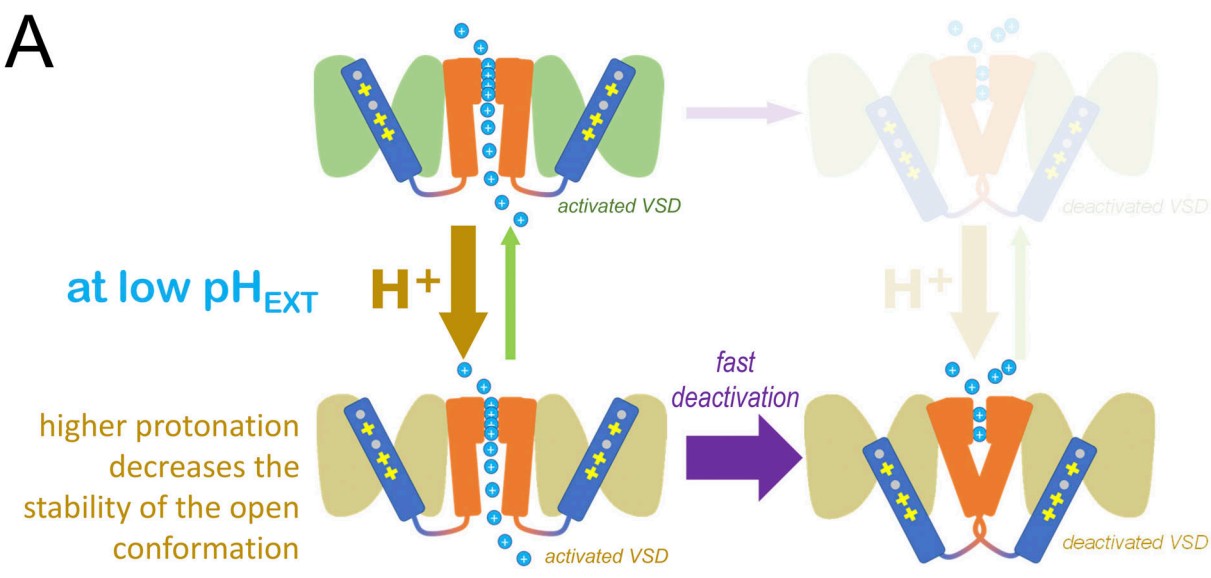

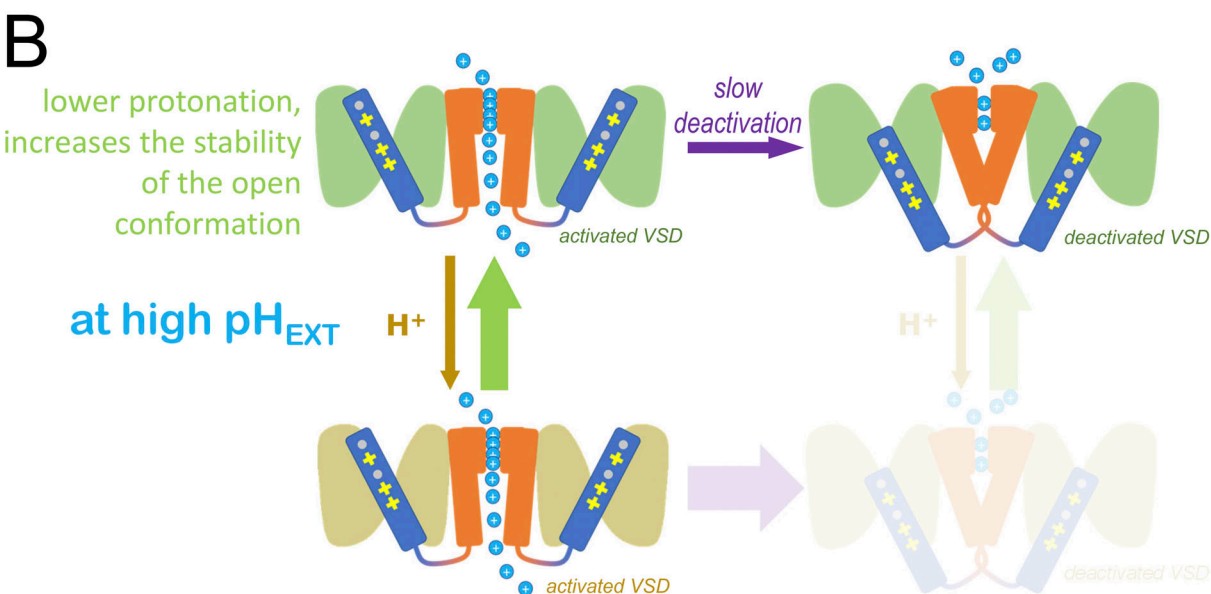

**Figure 11. Kinetic scheme providing an interpretation of the findings on the deactivation kinetics of the mutant R198Q.** Following activation, $K_V7.2$ channels tend to become resilient to close as they remain activated. This shows a reduction in the rate of deactivation, making it slower. The activated mutant R198Q becomes more stable as the channels remain activated by depolarization (Fig. 6 D). This happened at any $pH_{EXT}$. However, at low $pH_{EXT}$ deactivation was faster for any duration of the activating pulse than those observed at higher $pH_{EXT}$ (e.g., Fig. 6 D, $pH_{EXT}$ 6.0, blue circles, and $pH_{EXT}$ 8.0, purple squares, respectively). **(A and B)** This strongly suggests that the conformation of the highly protonated open channels (A) is less stable than that of the channels that are less protonated (B). Thus, at low $pH_{EXT}$, the channels readily deactivate, while at higher $pH_{EXT}$, the channels deactivate slowly.

suggested, as mentioned before, that the protein was not properly folded or simply was not functional. Identifying the reason for the lack of current also escapes the scope of this work. However, we observed that coexpression with $K_V7.3$ rendered robust $K^+$ currents in oocytes. This strongly suggested that the presence of the sister subunit was able to "mitigate" the effect of structural and functional deficiencies caused by the mutation. Furthermore, we observed that the resulting channels displayed a voltage dependence for activation that was shifted to more positive potentials with respect to the WT pair. Finally, we observed that the deactivation kinetics was unaltered by changes in $pH_{EXT}$. This was surprising to us, given the position of the residue in the VSD. We speculate that it is possible that the introduced histidine was always protonated in the range of $pH_{EXT}$ used in this study. However, we observed with the R198 mutant that the charge of the residue does not seem to be a determinant factor in the gained modulatory effect by $pH_{EXT}$. This suggests that conformational changes leading to the stabilization of the open conformation of the channel are highly $pH_{EXT}$ dependent and that the charge in position 198 disrupts that stabilized open conformation of the channel.

Figure 12. **Activation kinetic of the hetero-meric $K_V7.2/K_V7.3$ (WT) and $K_V7.2$-R198Q/ $K_V7.3$ (R198Q) channels. (A–D)** The *n*-powered two-exponential function was fitted to the activation phase of $K^+$ currents recording from oocytes expressing the WT $K_V7.2$ (A and B) and the mutant R198Q (C and D) coexpressed with $K_V7.3$. In both cases, the holding potential was set to −90 mV. The average fast (A and C) and slow (B and D) time constants were calculated from the parameters yielded by the fitting on individual experiments (WT: *n* = 9; R198Q: *n* = 8).

## A positively charged residue in position 198 disrupt $pH_{EXT}$-dependent open channel stabilization

At first, we thought that the lack of a positive charge in position 198 was facilitating the upward movement of the S4 segment at negative potentials simply by decreasing the electrical force driving the segment toward the intracellular side ("down"). In fact, titrating the introduced histidine in the mutant R198H made deactivation faster at low $pH_{EXT}$ than in a more alkaline environment. However, the mutant R198Q showed similar behavior, indicating that changes in the deactivation rate were associated with alteration in the stability of the open channels. In general, what we proposed is that the open-channel conformation becomes destabilized at acidic $pH_{EXT}$. Accordingly, the rate of deactivation increases at acidic $pH_{EXT}$ (Fig. 11 A). Conversely, the open conformation of the channels is further stabilized at alkaline $pH_{EXT}$, leading to a decrease in the rate of deactivation (Fig. 11 B).

What is the role of R198 in open channel stabilization? According to the work presented here, a positive charge in position 198 can disrupt the stabilization of the channel's open conformation. Increasing $pH_{EXT}$ caused the channels to become resilient to close as the deactivation became slower than that at lower $pH_{EXT}$. This is illustrated in the case of the mutant R198Q. Thus, the electrical properties of residue 198 are not sufficient to entirely explain the effect of $pH_{EXT}$ on the charge-neutralized mutants studied here.

## A final thought: Potential effect of the mutation R198Q on neuronal hyperexcitability

The $pH_{EXT}$-sensitive behavior of the mutant R198Q opens a new door to the understanding of its role in disease. Our initial question driving this project was: "How can a mutation that increases the activity of $K_V7.2$ channels produce hyperexcitability?" This apparent paradox was intriguing. We argue that

the changes in both voltage dependence and deactivation kinetics of the mutant R198Q could contribute to the prolongation of seizures/spasms. During high electrical activity, acidification of the extracellular environment in neuronal tissue can occur due to high electrical activity (Raimondo et al., 2015; Sulis Sato et al., 2017). In this situation, the low $pH_{EXT}$ will decrease the stability of the open mutant channel as well as shift its activation voltage dependence toward positive potential. These two actions would effectively reduce the basal $K^+$ conductance in the plasma membrane. Such a decrease could facilitate the triggering of action potentials. This would lead to further local acidification of the extracellular environment and eventually unchaining out-of-control activity, a seizure. In these terms, we propose that the mutation can be epileptogenic by enhancing excitability following local extracellular acidification. This hypothesis challenges the old notion that systemic acidosis ceases seizures, while alkalosis triggers them (Xiong et al., 2008; Cheng et al., 2021). In this case, we argue that the mutant channel is not responsible for the ictal stage of a seizure, instead the acidification drives a decrease in its activity and would further prolong high electrical activity.

Again, how can the mutation $K_V7.2$-R198Q lead to hyperactivity? Arguably, the number of open $K_V7$ channels open would not change through the course of a single neuronal action potential (nAP). This is because the channel's kinetics for activation and deactivation is slower than the time course of a nAP. In contrast, during a burst of nAP, the fraction of open $K_V7$ channels should increase. Consistent with this idea, $K_V7$ channels have been shown to contribute to the generation of after hyperpolarization potentials (Tzingounis and Nicoll, 2008; Kim et al., 2012; Larsson, 2013) among other channels (Cloues and Sather, 2003). In the case of the WT pair $K_V7.2/K_V7.3$, successive depolarizations during a burst of nAP would increase the number of $K_V7.2/K_V7.3$ channels that open, making the $K^+$ conductance more robust as nAP continue to fire. This effect on $K^+$ conductance will drive the termination of the burst. In contrast, for the mutant pair $K_V7.2$-R198Q/$K_V7.3$, the firing of successive nAP would lead to a decrease in $pH_{EXT}$ (Raimondo et al., 2015), causing an increase in the rate of deactivation, thereby making the $K^+$ conductance less robust. In addition to the increase of the deactivation rate, the mutant $K_V7.2$-R198Q/$K_V7.3$ activation becomes slower (Fig. 12, C and D) as the $pH_{EXT}$ decreases—this is not the case for the WT pair (Fig. 12, A and B). This suggests that the contribution of the mutant $K_V7.2$-R198Q/$K_V7.3$ channel to the $K^+$ conductance would decrease as $pH_{EXT}$ drops due to AP firing.

Another factor to consider is the contribution of $Na_V$ channels. Having a hyperpolarized membrane potential would recover $Na_V$ channels from resting state inactivation, making them available for activation, so increasing excitability. Using a simple model for AP in oocytes (Corbin-Leftwich et al., 2016, 2018), our preliminary data show that in the presence of the R198Q mutant, the maximum rate of depolarization during an AP is higher than those observed in the presence of the WT $K_V7.2$ channels. Furthermore, our preliminary data also show that the resting potential hyperpolarizes in the presence of the mutant channels and that the threshold potential ($V_{TH}$) for triggering

APs becomes more negative. This is different than what we reported on the effect of retigabine using a similar AP model in oocytes. In that case, retigabine hyperpolarized the membrane but left the $V_{TH}$ unaltered (Corbin-Leftwich et al., 2016). Thus, changing the $V_{TH}$ may be a key factor in the underlying mechanism for hyperexcitability caused by mutations like R198Q. Therefore, we hypothesize that the effect of the mutation R198Q goes beyond changes in the resting membrane potential and repolarization, having a direct impact on the activity of other channels (i.e., $Na_V$ channels), as the membrane potential couples their activity.

## Data availability
All data are available in the main text or the supplementary materials.

## Acknowledgments
Jeanne M. Nerbonne served as editor.

The authors thank Dr. Rene Barro-Soria for his comments.

The authors thank Extreme Science and Engineering Discovery Environment/Advanced Cyberinfrastructure Coordination Ecosystem: Services & Support (XSEDE/ACCESS) for providing support for computational work (Project: BIO220023 to C.A. Villalba-Galea). This study was funded by the National Institutes of Health (grant 1R21NS119854-01 to C.A. Villalba-Galea).

Author contributions: C.A. Villalba-Galea designed the study. B. Mehrdel performed the initial electrophysiological experiments and analyzed the initial data. C.A. Villalba-Galea performed electrophysiological experiments, analyzed the data, and wrote the initial and revised manuscripts.

Disclosures: The authors declare no competing interests exist.

Submitted: 19 October 2022

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

# Supplemental material

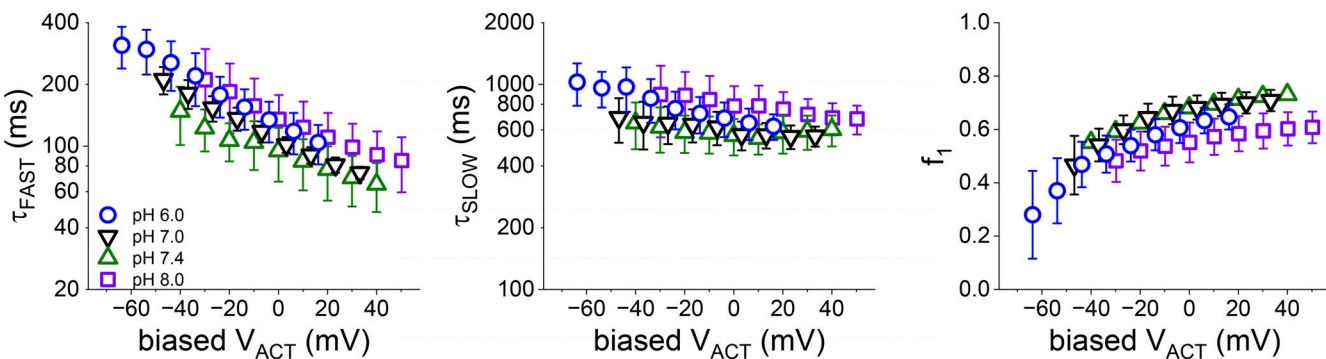

Figure S1.  **The mutant R198H displayed a shift in voltage dependence of activation that was about −17 mV per unit of pH$_{EXT}$.** If the change in voltage dependence is due to a purely electrostatic effect on the VSD, then the effect on the kinetic of activation is due to a local electrostatic bias. Accordingly, the plots in Fig. 3, G–I, were replotted changing the values of the X-axis at a rate of −17 m V/pH$_{EXT}$, using the V$_{ACT}$ values for the plot at pH$_{EXT}$ 7.4 as a reference, In doing so, we observed that the plots in Fig. 3, G–I, tend to overlap, strongly suggesting that the difference in the parameters t$_{FAST}$, t$_{SLOW}$, and f$_1$ are due to an electrostatic effect from the protonation.

