## [Peer Review File · The Journal of General Physiology]

Effect of a sensing charge mutation on the deactivation of KV7.2 channels

Baharak Mehrdel and Carlos Villalba-Galea

Corresponding Author(s): Carlos Villalba-Galea, University of the Pacific

Review Timeline:

Submission Date:	October 19, 2022
Editorial Decision:	November 8, 2022
Revision Received:	August 28, 2023
Editorial Decision:	September 16, 2023
Revision Received:	November 17, 2023
Editorial Decision:	November 30, 2023
Revision Received:	December 1, 2023
Editorial Decision:	December 5, 2023
Revision Received:	December 22, 2023

Editor: Jeanne Nerbonne

Transaction Report:

DOI: <https://doi.org/10.1085/jgp.202213284>

November 9, 2022

Dr. Carlos A Villalba-Galea
University of the Pacific
Physiology and Pharmacology
3601 Pacific Ave
Stockton, CA 95211

Re: 202213284

Dear Dr. Villalba-Galea,

Thank you for submitting your manuscript, titled "Effect of a sensing charge mutation on the deactivation of KV7.2 channels", to the Journal of General Physiology. Your manuscript has now been seen by three reviewers, whose comments are appended to this email. As you will see, each of the reviewers has raised several concerns about the design of some of the experiments, as well as about data presentation and interpretation, which will need to be addressed prior to further consideration of the manuscript at JGP. Additional concerns were also noted about the clarity of the text, the omission of detailed experimental protocols and supplemental data/figures, as well as the lack of statistical treatment of the experimental data presented.

The editors agree with the concerns identified by the reviewers. If you can address all of the reviewers' noted concerns, we would be happy to receive an extensively revised version of the present submission. The editors also wish to emphasize the need to improve the clarity of the presentation and suggest that this might be achieved by focusing on the major experimental findings and conclusions. In addition, it appears that several of the figures (for example, figures 1 and 2, figures 4 and 5, figures 7 and 8) could be combined and presented more concisely. Also, as presented figure 9 adds little, and this figure could be omitted without having a negative impact on the paper.

If you elect to revise and resubmit, please be aware that your resubmitted manuscript will be re-reviewed, most likely by three original referees, pending their availability. Based on the scope of the changes needed, we would anticipate that the revision process will take no longer than 6 months. If you find that you need additional, however, please do let us know and please also inform us as to what you think a realistic submission timeline will be. In addition, please do not hesitate to contact me (via the editorial office) if you feel that a discussion of the reviewers' and editors' comments would be helpful.

Please submit your revised manuscript via the link below along with a point-by-point letter that details your responses to the editors' and reviewers' comments, as well as a copy of the text with alterations highlighted (boldfaced or underlined). If the article is eventually accepted, it would include a 'revised date' as well as submitted and accepted dates. If we do not receive the revised manuscript within one year, we will regard the article as having been withdrawn. We would be willing to receive a revision of the manuscript at a later time, but the manuscript will then be treated as a new submission, with a new manuscript number.

Please pay particular attention to recent changes to our instructions to authors in sections: Data presentation, Blinding and randomization and Statistical analysis, under Materials and Methods, as shown here: <https://rupress.org/jgp/pages/submission-guidelines#prepare>. Re-review will be contingent on inclusion of the required information (including for data added during revision) and demonstration of the experimental reproducibility of the results (i.e., all experimental data verified in at least 2 independent experiments).

When revising your manuscript, please be sure it is a double-spaced MS Word file and that it includes editable tables, if appropriate.

Please submit your revised manuscript via this link:
Link Not Available

Thank you for submitting this manuscript to JGP.

Sincerely,

Jeanne Nerbonne, Ph.D.
On behalf of Journal of General Physiology

Journal of General Physiology's mission is to publish mechanistic and quantitative molecular and cellular physiology of the highest quality; to provide a best-in-class author experience; and to nurture future generations of independent researchers.

Reviewer #1 (Comments to the Authors):

This manuscript tries to explain why mutations of R198 in Kv7.2 channels at the top of S4 can cause epilepsy. They show that mutations R198H shifts the voltage dependence and that this shift is pH dependent. In addition, the deactivation kinetics is very pH dependent in this mutation, while activation kinetics is less pH dependent. In contrast, the wt Kv7.2 is not very pH dependent. This suggests that the pH effects are due to protonation/deprotonation of R198H. Surprisingly, R198Q displays the same type of pH dependence. Co-expression of Kv7.3 and R198 mutations of Kv7.2 display the same type of pH dependence as the Kv7.2 mutations in homomeric channels, suggesting the effects are due to mechanism internal to the Kv7.s subunits. The authors present a somewhat confusing model how the pH affects deactivation in Kv7.2. The manuscript would be greatly strengthened by identification of the protonated residues underlying the effects. Or at least, some effort displayed at identifying the residues by mutating a couple of protonatable residues in the Kv7.2 R198Q background. Their suggestion for how the R198Q mutation cause epilepsy through effects by acidosis is imaginative and increases the impact of the paper.

Specific Comments.

1. Identification of the protonated residue would greatly strengthen the paper. One idea is that the residue that gets protonated is E130 in S2 which R198 has been proposed to interact with. What about mutating this residue in R198Q background to see whether it removes the pH effect?
2. The authors call the fitting functions Fermi-Dirac whereas most people in the field call the functions Boltzmann curves. Fermi-Dirac is usually used in Quantum Mechanics. Why introduce it here?
3. Line 135-138. These arguments are not clearly explained.
4. Line 305-311. These conclusions and arguments could be better explained. For example, charges do not have to move from one side to the other side to contribute to gating. The field does not drive the movement. Instead thermal motion drives the movement, the electrical field just bias the average movement. A residue contributes more if it is actually outside the hydrophobic core, moving from one side to the other side.
5. The model proposed is not easy to understand. What do A3 and @3 in figure represent?

Reviewer #2 (Comments to the Authors):

The voltage-gated potassium channel Kv7.2 plays an important role in neurons and contributes to regulating neuronal excitability. Loss-of-function mutations in the gene encoding for Kv7.2 is linked to inherited forms of epilepsy. Hence, understanding how Kv7.2 function is impacted by mutations and how this relates to the clinical phenotype is of great interest. One puzzling finding for Kv7.2 and other Kv channels is that also apparent gain-of-function mutations are linked to hyperexcitability conditions like epilepsy. In this study, Mehrdel and Villalba-Galea perform detailed biophysical studies of one such mutation to gain insights into the mechanism(s) underlying channel dysfunction. They found that charge-neutralizing mutations of the outermost gating arginine in S4, R198, equips the channel with a more prominent sensitivity to changes in extracellular pH, compared to the WT channel. In particular, the deactivation kinetics was affected by extracellular pH with clear speeding up of the kinetics in response to more acidic pH and slowing down of kinetics in response to less acidic pH. Moreover, the voltage dependence of mutant channel opening was clearly shifted in response to altered external pH. The authors propose that alkaline pH alters protonation of an unknown element at the extracellular face of the channel, which triggers open channel stabilization. Moreover, the authors propose that a positive charge at position 198 largely overcomes this stabilization, allowing the channel to deactivate relatively readily. In contrast, loss of a positive charge at position 198, as in R198H/Q mutants, impairs the ability of the channel to deactivate, leading to slower deactivation kinetics.

The authors describe several interesting findings in this study, submitted in the format of a Communication. Their detailed biophysical characterization of the behaviour of R198 mutants under different conditions will be of interest in the ion channel research field and provides important insights into the complex interplay between inherited mutations and physiological conditions such as external pH (which may change during hyperexcitability conditions). The experiments seem to be of high quality and carefully designed. However, there are some aspects regarding data analysis and presentation, which the authors need to address. Moreover, the authors need to strengthen some of their conclusions by clarifying the effect of mutations on different steps during channel deactivation and by more clearly describing whether the mutants truly cause a loss-of-function phenotype compared to WT. Please refer to the detailed comments below.

General comments:

1. Some critical information is missing in the methods section. For row 124-128, could the authors please comment upon what was the reason for fitting a double Fermi-Dirac function to achieve a weighted $V_{1/2}$? Also, please add to the methods section information about the equation and how the fit was done. In Figure 2B and C, it looks like the shift in $V_{1/2}$ induced by altered external pH is not linear, as the $V_{1/2}$ of R198H is roughly similar at pH 7.4 and pH 8. Yet, the authors fit the data using a linear regression in Figure 2C. Please comment upon possible limitations related to this. Also, related to row 140-144, please add to the methods section information about the equation and how the fit was done.
2. One of the major focuses of the Discussion is the authors proposed model of channel deactivation happening in at least two

steps and that the loss of R1 may impair the second step. Yet, if I understand the data in Figure 5 and 6 correct, the authors present only the weighted average time constant. What is the reason for reporting only the weighted average time constant for deactivation, whereas both the fast and slow components are presented for activation (Figure 3)? Given the emphasis of putative effects of mutations and protonation status on the different deactivation steps in the Discussion, it seems reasonable that the authors would support their model with more detailed experimental data for deactivation kinetics. For instance, would it be useful to report on the effect of mutations and altered external pH on the fast and slow component of channel deactivation? Or are there alternative ways of quantifying the proposed effect on step 1 and 2, respectively, of channel deactivation? In relation to this, the small "pseudo plateau" (row 318) of the deactivation is not very apparent to me. Please indicate the plateau in Figure 5A.

3. Another main conclusion from this study is that the shifted $V_{1/2}$ and faster deactivation kinetics of the R198H/Q mutants triggered by extracellular acidification leads to channel loss-of-function, which could explain the link to hyperexcitability and epilepsy (e.g. row 93-94 and 363-364). Although absolute numbers for $V_{1/2}$ and channel kinetics are not readily compared from the presented data, it looks to me that the deactivation kinetics of the mutants are always slower than for WT and that $V_{1/2}$ of the mutants are always more negative than for WT. For instance, even at pH 6 the R198H mutant display slower deactivation kinetics than WT at pH 6 (Figure 1) and $V_{1/2}$ of the mutant is still more negative than for WT (Figure 2A-B). Hence, compared to WT, it seems to me that the mutants display a gain-of-function phenotype at acidic pH also for these parameters, when compared to the WT channel. This would rather protect from hyperexcitability, similar to what is observed for retigabine (row 76-78). Could the authors please comment on this and correct me if I am wrong?

4. A general comment is that statistics is largely missing in the present version of the manuscript. The only exception is Figure 7, for which P values are included but not explained in the figure legend.

5. Row 281-283: The authors refer to data in Figure S1. I could not find this figure.

Specific comments:

Row 102-103: "...was (Fig. 1A, B) was seemingly..." Please remove one "was"

Row 148: Please refer to panel 3C at the end of the sentence "...for the mutant R198H."

Row 189-191: "Consistent to previous reports, this suggested that the changes in pHEXT were due to a transition to a more stable conductive/activated conformation...". Please add references to the mentioned previous reports. Also, should "pHEXT" perhaps be "deactivation kinetics"?

Row 202: "...mutations..." should be "...mutation..."

Row 206: "... (Fig. 7B)." should be 6B. Also, please refer to panel 6C and D at appropriate places in the subsequent section.

Row 250: "...lost-of-function..." should be "...loss-of-function..."

Row 258: "A recent study...". Please add reference to the mentioned previous study.

Row 260: Please remove "the showed".

Row 284: Change "has" to "have"?

Row 292: Should be "Figs. 5 & 6"?

Row 293-295 "This is consistent with previous observations...". Please add references to the mentioned previous studies.

Row 308: "...residues in..." Add "is"?

Row 313, 326, 327: Please correct multiple grammatical errors in these sentences.

Row 329-331: This sentence is hard to understand. Could the authors please clarify what they mean by "monotonicity" and "latter parameter" in this context?

Methods: Please describe how the high and low pH, respectively, was set.

Figure 6 legend: Please indicate the number of recordings (n).

Figure 9 legend: Please clarify the numerical information related to step 2 in the upper row.

Reviewer #3 (Comments to the Authors):

In this manuscript, Mehrdel and Villalba-Galea characterize the activation and deactivation kinetics of KCNQ2 as a result of change in the charge of the R198 residue in the voltage-sensing domain. They introduced R198H and R198Q mutations into KCNQ2 channels so that they could alter the charge on this residue based on the pH of the external solutions. They then fit the kinetics of current activation and deactivation in various experimental states to determine the time constants (τ). They have concluded that while external pH has modest effect on the activation kinetics, it significantly alters deactivation kinetics of the channel, especially in the absence of positive charge in residue 198.

Here, the experiments are well thought of and well carried out. Figures are easy to understand. However, part of the text can be challenging to follow at times. Some of the conclusion statements or conjectures at the end of the paragraphs are not always well supported by experiment data or cited literature.

Major Comments

#1 In Figure 7, you mentioned the effect of Kv7.3 on Kv7.2. Did you control for the ratio of the expression between the two subunits? One could wonder that more Kv7.3 subunits may alter the overall effect of Kv7.2/7.3 channels.

#2 In Figure 5B, there are large error bars in deactivation time constant of WT. Was that truly due to experimental variation or was it because the function used to fit the kinetics was not sufficient to "catch" or describe the multi-phasic nature of the kinetics?

#3 Lines 319-323. There is no experimental evidence to support the claim that "charge of residue 198 is critical for the second phase of deactivation." Unlike activation time constants (τ_{fast} and τ_{slow}), they only fit one τ for deactivation. See comment #2.

#4 One of the two conclusions in the abstract was not well described in the discussion section. Although the second conclusion about deactivation is very well described in the conclusion, the first conclusion about protonation status of the channel's extracellular face is less well discussed and/or supported by the experimental data. Please add or clarify this discussion point.

Minor Comments

#1 Lines 261-267 discuss the relationship between steepness of the current-vs-voltage curve and the number of sensing charges. Here, the authors contradicted themselves whether they could imply the role of the R198 residue as a sensing charge. Please rephrase the discussion.

#2 There are multiple typos, grammatical errors, missing figure citation, and missing literature citation. Please proofread carefully prior to re-submission. Below are only some of the examples and did not include all the errors found in the manuscript.

- Supplementary Figure #1 is mentioned in the manuscript but was not available to review in the submission.
- Line 128 mentions 'weighted $V_{1/2}$.' This weighted $V_{1/2}$ equation was not included/described in the method section.
- Line 141: please write out a full equation of the "n-powered two-exponential function" used.
- Lines 164-166 contain a run-in sentence which is difficult to understand. Please rephrase.
- Lines 169-170: "...we proceeded to study the deactivation kinetics as a function of activation." What do you mean by 'activation'? Which particular features of the activation kinetics are you referring to?
- Line 410: where does f_2 come from? Is there another equation that f_2 is a part of?
- Lines 189-191: "Consistent with previous reports, this suggested that changes in pHEXT were due to a transition..." There is no literature cited here.
- Line 206: when you mention figure 7B, did you mean 6B? Figure 6D was also not described in the main text.
- Lines 258-261: There is a missing citation again.

#3 I recommend citing PMID: 35642783 (Edmond et al, eLife, 2022) which describes the effect of R198Q on the S4 movement in KCNQ2

Reviewer 1

This manuscript tries to explain why mutations of R198 in Kv7.2 channels at the top of S4 can cause epilepsy. They show that mutations R198H shifts the voltage dependence and that this shift is pH dependent. In addition, the deactivation kinetics is very pH dependent in this mutation, while activation kinetics is less pH dependent. In contrast, the wt Kv7.2 is not very pH dependent. This suggests that the pH effects are due to protonation/deprotonation of R198H. Surprisingly, R198Q displays the same type of pH dependence. Co-expression of Kv7.3 and R198 mutations of Kv7.2 display the same type of pH dependence as the Kv7.2 mutations in homomeric channels, suggesting the effects are due to mechanism internal to the Kv7.s subunits. The authors present a somewhat confusing model how the pH affects deactivation in Kv7.2. The manuscript would be greatly strengthened by identification of the protonated residues underlying the effects. Or at least, some effort displayed at identifying the residues by mutating a couple of protonatable residues in the Kv7.2R198Q background. Their suggestion for how the R198Q mutation cause epilepsy through effects by acidosis is imaginative and increases the impact of the paper.

We agree with the reviewer about the comment on the the complexity of the model proposed. To address this issue, the number of states has been decreased from 6 to 4, hoping that will make it clearer. On the same line of thought, the proposal of a “at-least-two-stage” process for activated has been removed from the manuscript and will be address in future research.

Specific Comments.

1. Identification of the protonated residue would greatly strengthen the paper. One idea is that the residue that gets protonated is E130 in S2 which R198 has been proposed to interact with. What about mutating this residue in R198Q background to see whether it removes the pH effect?

We found this to be a great idea, so we proceeded to implement it directly. We reasoned that the best bet would have been the mutant E130H because it would allow us to play with the charge of the residue. However, we found not increased sensitivity with this mutant. Also, our preliminary data show no effect with the mutant E130Q either (data no shown). We considered studying a double mutant (e.g.: E130H-R198Q) but left it for future research work.

2. The authors call the fitting functions Fermi-Dirac whereas most people in the field call the functions Boltzmann curves. Fermi-Dirac is usually used in Quantum Mechanics. Why introduce it here?

Thank you for bringing this to our attention. This is not the first introduction to this issue. I have added an explanation for the use of the name Fermi-Dirac between line 137 and 140.

3. Line 135-138. These arguments are not clearly explained.

Thank you for pointing this out. The statement has been revised and is now located between lines 152 and 156. It reads as follows “...In other words, changes in the protonation status of the introduced histidine only seemed to change the electrical field around the VSD, rather than the ability for the mutant channel to sense the electrical field across the membrane. Thus, the charge of the histidine seems to only bias the electrical field at the resting/deactivated state of the VSD...”

4. Line 305-311. These conclusions and arguments could be better explained. For example, charges do not have to move 3 from one side to the other side to contribute to gating. The field does not drive the movement. Instead thermal motion drives the movement, the electrical field just bias the average movement. A residue contributes more if it is actually outside the hydrophobic core, moving from one side to the other side.

These arguments have been removed for the sake of clarity.

5. The model proposed is not easy to understand. What do A3 and @3 in figure represent?

The model has been extensively revised and simplified. See Figure 11.

Reviewer 2

The voltage-gated potassium channel Kv7.2 plays an important role in neurons and contributes to regulating neuronal excitability. Loss-of-function mutations in the gene encoding for Kv7.2 is linked to inherited forms of epilepsy. Hence, understanding how Kv7.2 function is impacted by mutations and how this relates to the clinical phenotype is of great interest. One puzzling finding for Kv7.2 and other Kv channels is that also apparent gain-of-function mutations are linked to hyperexcitability conditions like epilepsy. In this study, Mehrdel and Villalba-Galea perform detailed biophysical studies of one such mutation to gain insights into the mechanism(s) underlying channel dysfunction. They found that charge-neutralizing mutations of the outermost gating arginine in S4, R198, equips the channel with a more prominent sensitivity to changes in extracellular pH, compared to the WT channel. In particular, the deactivation kinetics was affected by extracellular pH with clear speeding up of the kinetics in response to more acidic pH and slowing down of kinetics in response to less acidic pH. Moreover, the voltage dependence of mutant channel opening was clearly shifted in response to altered external pH. The authors propose that alkaline pH alters protonation of an unknown element at the extracellular face of the channel, which triggers open channel stabilization. Moreover, the authors propose that a positive charge at position 198 largely overcomes this stabilization, allowing the channel to deactivate relatively readily. In contrast, loss of a positive charge at position 198, as in R198H/Q mutants, impairs the ability of the channel to deactivate, leading to slower deactivation kinetics. The authors describe several interesting findings in this study, submitted in the format of a Communication. Their detailed biophysical characterization of the behaviour of R198 mutants under different conditions will be of interest in the ion channel research field and provides important insights into the complex interplay between inherited mutations and physiological conditions such as external pH (which may change during hyperexcitability conditions). The experiments seem to be of high quality and carefully designed. However, there are some aspects regarding data analysis and presentation, which the authors need to address. Moreover, the authors need to strengthen some of their conclusions by clarifying the effect of mutations on different steps during channel deactivation and by more clearly describing whether the mutants truly cause a loss-of-function phenotype compared to WT. Please refer to the detailed comments below.

General comments:

1. Some critical information is missing in the methods section. For row 124-128, could the authors please comment upon what was the reason for fitting a double Fermi-Dirac function to achieve a weighted $V_{1/2}$? Also, please add to the methods section information about the equation and how the fit was done. In Figure 2B and C, it looks like the shift in $V_{1/2}$ induced by altered external pH is not linear, as the $V_{1/2}$ of R198H is roughly similar at pH 7.4 and pH 8. Yet, the authors fit the data using a linear regression in Figure 2C. Please comment upon possible limitations related to this. Also, related to row 140-144, please add to the methods section information about the equation and how the fit was done.

Thank you to the reviewer for pointing out these deficiencies in the manuscript. In addressing them, we have introduced two additional sub-section under Methods, describing the equations in question.

Regarding the use of a linear function to describe the pH dependence of the $V_{1/2}$, we did such for the sake of simplicity. Using any other function – like a titration function of any kind – would be too speculative in our view. So, we chose to use a linear relationship. We agree that the $V_{1/2}$ for pH_{EXT} 7.4 and 8.0 are similar. Yet, we preferred to characterize the overall effect with a simplistic linear function.

2. One of the major focuses of the Discussion is the authors proposed model of channel deactivation happening in at least two steps and that the loss of R1 may impair the second step. Yet, if I understand the data in Figure 5 and 6 correct, the authors present only the weighted average time constant. What is the reason for reporting only the weighted average time constant for deactivation, whereas both the fast and slow components are presented for activation (Figure 3)? Given the emphasis of putative effects of mutations and protonation status on the different deactivation steps in the Discussion, it seems reasonable that the authors would support their model with more detailed experimental data for deactivation kinetics. For instance, would it be useful to report on the effect of mutations and altered external pH on the fast and slow component of channel deactivation? Or are there alternative ways of quantifying the proposed effect on step 1 and 2, respectively, of channel deactivation? In relation to this, the small "pseudo plateau" (row 318) of the deactivation is not very apparent to me. Please indicate the plateau in Figure 5A.

We agree with the reviewer in what we understand is a prematurely made claim on the multi-step nature of the deactivation. So, responding to this comment and a similar one from reviewer 1, we have opted to abandon – for now – the idea that deactivation takes place in at least two stages. Our revised scheme (Figure 11) provides a much simpler explanation of our results.

3. Another main conclusion from this study is that the shifted $V_{1/2}$ and faster deactivation kinetics of the R198H/Q mutants triggered by extracellular acidification leads to channel loss-of-function, which could explain the link to hyperexcitability and epilepsy (e.g. row 93-94 and 363-364). Although absolute numbers for $V_{1/2}$ and channel kinetics are not readily compared from the presented data, it looks to me that the deactivation kinetics of the mutants are always slower than for WT and that $V_{1/2}$ of the mutants are always more negative than for WT. For instance, even at pH 6 the R198H mutant display slower deactivation kinetics than WT at pH 6 (Figure 1) and $V_{1/2}$ of the mutant is still more negative than for WT (Figure 2A-B). Hence, compared to WT, it seems to me that the mutants display a gain-of-function phenotype at acidic pH also for these parameters, when compared to the WT channel. This would rather protect from hyperexcitability, similar to what is observed for retigabine (row 76-78). Could the authors please comment on this and correct me if I am wrong?

Thank you to the reviewer for this insightful question. Through the course of a single neuronal action potential (nAP), it can be argued that the number of open K_{V7} channels open would not change, as they have a slower kinetics for activation and deactivation compared to the time course of a nAP. In contrast, during rapid firing of more than one nAP, it is likely a fraction of open K_{V7} channels would increase. In fact, K_{V7} channels have been shown to the generation of afterhyperpolarization potentials (AHP). Following this idea, let's consider the WT pair $K_{V7.2}/K_{V7.3}$. During successive AP, the number of open $K_{V7.2}/K_{V7.3}$ channels increases, while the number of depolarizing channels (e.g.: Na_V channels) recovering from inactivation decreases, reaching a point at which a prologue AHP is observed. During AHP, $K_{V7.2}/K_{V7.3}$ channels would close as the membrane potential is made more negative than the resting potential for few tens of milliseconds. Eventually, the membrane returns to its resting potential and a small fraction of $K_{V7.2}/K_{V7.3}$ channels will be opened. Now, let us consider a similar scenario, but with the mutant $K_{V7.2}\text{-R198Q}/K_{V7.3}$ channels. In this case, triggering nAP is likely

harder as the resting K^+ conductance is expected to be higher. However, after rapid firing, the pH_{EXT} would decrease. For the WT channel, this was not an issue, because its voltage dependence and kinetic are fairly pH_{EXT} insensitive. For the mutant channel, the voltage dependence shifts closer to that of the WT, but not the kinetic of activation. For the mutant $K_V7.2-R198Q/K_V7.3$, activation becomes slower (Fig. 12C,D) than that of the WT pair (Fig. 12A,B). In this latter case, we propose that the development of the AHP is delayed as the activation of the K_V7 current is delayed, leading to longer AP firing events.

4. A general comment is that statistics is largely missing in the present version of the manuscript. The only exception is Figure 7, for which P values are included but not explained in the figure legend.

Thank you to the reviewer for pointing this out. We have addressed this throughout the manuscript.

5. Row 281-283: The authors refer to data in Figure S1. I could not find this figure.

We apologize for this omission. We have incorporated Figure S1.

Specific comments:

Row 102-103: "...was (Fig. 1A, B) was seemingly..." Please remove one "was"

We have addressed this.

Row 148: Please refer to panel 3C at the end of the sentence "...for the mutant R198H."

We have addressed this.

Row 189-191: "Consistent to previous reports, this suggested that the changes in pH_{EXT} were due to a transition to a more stable conductive/activated conformation...". Please add references to the mentioned previous reports.

Also, should " pH_{EXT} " perhaps be "deactivation kinetics"?

We have addressed this.

Row 202: "...mutations..." should be "...mutation..."

We have addressed this.

Row 206: "... (Fig. 7B)." should be 6B. Also, please refer to panel 6C and D at appropriate places in the subsequent section.

We have addressed this.

Row 250: "...lost-of-function..." should be "...loss-of-function..."

We have addressed this.

Row 258: "A recent study...". Please add reference to the mentioned previous study.

We have addressed this.

Row 260: Please remove "the showed".

We have addressed this.

Row 284: Change "has" to "have"?

We have addressed this.

Row 292: Should be "Figs. 5 & 6"?

We have addressed this.

Row 293-295 "This is consistent with previous observations...". Please add references to the mentioned previous studies.

We have addressed this.

Row 308: "...residues in..." Add "is"?

We have addressed this.

Row 313, 326, 327: Please correct multiple grammatical errors in these sentences.

We have addressed this.

Row 329-331: This sentence is hard to understand. Could the authors please clarify what they mean by "monotonicity" and "latter parameter" in this context?

We have addressed this.

Methods: Please describe how the high and low pH, respectively, was set.

We have addressed this.

Figure 6 legend: Please indicate the number of recordings (n).

We have addressed this.

Figure 9 legend: Please clarify the numerical information related to step 2 in the upper row.

We have addressed this.

Reviewer 3

In this manuscript, Mehrdel and Villalba-Galea characterize the activation and deactivation kinetics of KCNQ2 as a result of change in the charge of the R198 residue in the voltage-sensing domain. They introduced R198H and R198Q mutations into KCNQ2 channels so that they could alter the charge on this residue based on the pH of the external solutions. They then fit the kinetics of current activation and deactivation in various experimental states to determine the time constants (τ). They have concluded that while external pH has modest effect on the activation kinetics, it significantly alters deactivation kinetics of the channel, especially in the absence of positive charge in residue 198. Here, the experiments are well thought of and well carried out. Figures are easy to understand. However, part of the text can be challenging to follow at times. Some of the conclusion statements or conjectures at the end of the paragraphs are not always well supported by experiment data or cited literature.

Major Comments

#1 In Figure 7, you mentioned the effect of Kv7.3 on Kv7.2. Did you control for the ratio of the expression between the two subunits? One could wonder that more Kv7.3 subunits may alter the overall effect of Kv7.2/7.3 channels. It is not clear for me what the reviewer refers to when say "control for the ratio of the expression". We co-expressed both subunits in equal concentration.

#2 In Figure 5B, there are large error bars in deactivation time constant of WT. Was that truly due to experimental variation or was it because the function used to fit the kinetics was not sufficient to "catch" or describe the multi-phasic nature of the kinetics?

Thank you for this question, we only observed that at $\text{pH}_{\text{EXT}} 7.0$. We found that it was the way the data came out and we decided not to "cherry-picked" the data. We however repeated the experiments with the mutant R198H because we found a way to improve the quality of the recordings. In a few words, we noticed that holding the membrane at -90 mV or -120 mV was causing the recordings to become unstable. Probably for high accumulation of K^+ on the extracellular side of the membrane. So, we instead, changed the holding potential to -50 mV, which is close to the theoretical reversal potential for K^+ for concentrations used in our

extracellular and intracellular recording solutions, 12 mM and 100 mM, respectively. Using this approach, the variability was unambiguously reduced.

#3 Lines 319-323. There is no experimental evidence to support the claim that "charge of residue 198 is critical for the second phase of deactivation." Unlike activation time constants (τ_{fast} and τ_{slow}), they only fit one τ for deactivation.

Thanks for this comment. We have abandoned this line of reasoning for this manuscript. We agree with the reviewer that the evidence was not there.

See comment #2.

#4 One of the two conclusions in the abstract was not well described in the discussion section. Although the second conclusion about deactivation is very well described in the conclusion, the first conclusion about protonation status of the channel's extracellular face is less well discussed and/or supported by the experimental data. Please add or clarify this discussion point.

We have extensively revised the manuscript. We hope to have addressed this.

Minor Comments

#1 Lines 261-267 discuss the relationship between steepness of the current-vs-voltage curve and the number of sensing charges. Here, the authors contradicted themselves whether they could imply the role of the R198 residue as a sensing charge. Please rephrase the discussion.

We have addressed this.

#2 There are multiple typos, grammatical errors, missing figure citation, and missing literature citation. Please proofread carefully prior to re-submission. Below are only some of the examples and did not include all the errors found in the manuscript.

We apologize for this. We have made a big effort towards addressing this.

- Supplementary Figure #1 is mentioned in the manuscript but was not available to review in the submission.

We have addressed this. We apologize for the omission.

- Line 128 mentions 'weighted $V_{1/2}$.' This weighted $V_{1/2}$ equation was not included/described in the method section.

We have addressed this.

- Line 141: please write out a full equation of the "n-powered two-exponential function" used.

We have addressed this.

- Lines 164-166 contain a run-in sentence which is difficult to understand. Please rephrase.

We have addressed this. We apologize for that.

- Lines 169-170: "...we proceeded to study the deactivation kinetics as a function of activation." What do you mean by 'activation'? Which particular features of the activation kinetics are you referring to?

We referred to the duration of the activating pulse to +40 mV and the weighted time constant of deactivation. We have shown in two previous publications that there is a non-linear relationship between these two.

- Line 410: where does f_2 come from? Is there another equation that f_2 is a part of?

We have addressed this. This shouldn't have been there. Sorry for the confusion.

- Lines 189-191: "Consistent with previous reports, this suggested that changes in pHEXT were due to a transition..." There is no literature cited here.

We have addressed this.

- Line 206: when you mention figure 7B, did you mean 6B? Figure 6D was also not described in the main text.
Sorry for the confusion. This has been revised.

- Lines 258-261: There is a missing citation again.
Consistent with previous reports, this
Sorry for the omission. This has been revised.

#3 I recommend citing PMID: 35642783 (Edmond et al, eLife, 2022) which describes the effect of R198Q on the S4 movement in KCNQ2
Thank you for the suggestion.

September 18, 2023

Dr. Carlos A Villalba-Galea
University of the Pacific
Physiology and Pharmacology
3601 Pacific Ave
Stockton, CA 95211

Re: 202213284R1

Dear Dr. Villalba-Galea,

We have received the reviews of your revised manuscript, titled "Effect of a sensing charge mutation on the deactivation of KV7.2 channels"; the reviewers' comments are appended below. As you will see, although Reviewer #1 is satisfied with the revised manuscript, Reviewers #2 and #3 have raised a several issues about the data presented, and both again request that detailed statistical analyses of the data be presented throughout and that you carefully review the manuscript for clarity and readability. The editors concur with the reviewers' comments and recommendations.

We would be pleased to receive a suitably revised manuscript that addresses the concerns noted in the critiques and summarized above. While the editors agree with reviewer #1 and #2 that "the E130 mutations would have been more informative if they were made in the background of R198Q instead of the wild-type background", we don't require that these additional experiments be completed. Based on the scope of the requested changes, we would anticipate that the revision process will take no longer than 2 months. If, however, you need additional time, please let us know. In addition, please do not hesitate to contact me (via the editorial office) if you feel that a discussion of the reviewers' and editors' comments would be helpful.

Please submit your revised manuscript via the link below along with a point-by-point letter that details your responses to the editors' and reviewers' comments, as well as a copy of the text with alterations highlighted (boldfaced or underlined).

When preparing your revised manuscript, please also pay particular attention to recent changes to our instructions to authors in the following sections: Data presentation, Blinding and randomization and Statistical analysis, under Materials and Methods, as shown here: <https://rupress.org/jgp/pages/submission-guidelines#prepare>. Re-review will be contingent on inclusion of the required information (including for data added during revision) and demonstration of the experimental reproducibility of the results. Also, to improve the reproducibility of published content, we have partnered with SciScore. Authors are prompted in eJP to copy and paste the Materials and Methods section of their manuscript for a SciScore assessment when submitting their revised manuscript. Authors are encouraged (not required) to further revise their Materials and Methods if the SciScore is below 4. More information can be found here: <https://rupress.org/jgp/pages/submission-guidelines#sciscore>

When revising your manuscript, please also be sure that it is a double-spaced MS Word file and that it includes editable tables, if appropriate.

Please submit your revised manuscript via this link:
Link Not Available

Thank you for the opportunity to consider your manuscript.

Sincerely,

Jeanne Nerbonne, Ph.D.
On behalf of the Journal of General Physiology

Journal of General Physiology's mission is to publish mechanistic and quantitative molecular and cellular physiology of the highest quality; to provide a best-in-class author experience; and to nurture future generations of independent researchers.

Reviewer #1 (Comments to the Authors):

The authors have responded well to my comments. However, the E130 mutations would have been more informative if they were made in the background of R198Q instead of wt background.

Reviewer #2 (Comments to the Authors):

The revised manuscript is much improved. For instance, the authors have extensively expanded the methods section to add previously missing information and have revised the discussion section and model to better align with their experimental observations of deactivation kinetics. However, some sections are still difficult to follow and some of my previous comments have not been fully addressed. Further adjustments would improve the quality of the work and make the conclusions easier to understand:

1. I find the authors response to my previous comment #3 (how the mutation induces hyperexcitability) and their associated discussion (row 461-482) difficult to follow. If I understand it correctly, the authors conclude that the disease phenotype of the R198Q mutant relates to the role of the Kv7.2/Kv7.3 channel in establishing the AHP, and that the slower activation kinetics of the mutant delay establishment of the AHP. I do not fully understand how they come to the conclusion that this is the anticipated functional effect of the mutant during neuronal firing given the mutant's complex behavior compared to WT (shifted V1/2, altered activation kinetics, dramatically altered deactivation kinetics). Additionally, how does this relate to the conclusion made on row 100-106, which rather points to the altered deactivation kinetics as the underlying cause?
2. The authors responded to my previous comment #4 (missing statistics) that they had addressed this throughout the manuscript. However, I cannot find statistics supporting any of the authors claims. For instance, no statistics is given to support claims of shifts/no shifts in V1/2, or changes to kinetics. Several of the authors claims are convincing even without statistics, but other claims would be strengthened by inclusion of appropriate statistical calculations (such as T tests or ANOVA's). On row 526, the authors state that they use T tests to calculate statistics for time constraints. Where is the outcome of this statistical analysis reported?
3. Good that the authors now include information about the Fermi-Dirac fit in the methods section. What is the reason for not including the outcome of the fit, in the form of lines, in the Itail-Vact plots? Without the fits, it is hard for the reader to judge how well the data is fitted.
4. For the structural model included in the revision (row 279-281), please include appropriate information in the methods section.
5. I liked the suggestion made by Reviewer 1 to test mutating residue E130 in a R198Q background, to see if this removes the pH effect. However, as far as I understand, the authors tested a E130H mutation in the WT background. What was the reason for this? Given that the WT channel shows limited pH effects, it is not clear to me what the authors expected from the E130H mutation alone. Also, because of the poor expression/function of the E130H mutant, an alternative would be to test the E130C or E130R mutations in a R198Q background, as the E130C and E130R mutants have been previously successfully studied using electrophysiology (Soldovieri et al., 2019, doi: 10.3390/ijms20143382).
6. Row 461-462: "If the voltage-dependence for activation is similar for Kv7.2/Kv7.3 and for Kv7.2-R198Q/Kv7.3..." This sentence can be misinterpreted. The voltage dependence is clearly different for the WT and mutant heteromers at physiological pH (pH 7.4), as shown in Figure 7. However, V1/2 of mutants approach that of WT during acidification. The sentence can be re-phrased to clarify this.
7. Figure 7 legend: The figure legend does not match the figure itself, which has panel A-H.
8. There are many grammatical errors and unclear language throughout the text, which requires proofreading.

Reviewer #3 (Comments to the Authors):

In this revised manuscript, Mehrdel and Villalba-Galea have added a new experiment, rephrased the discussion conclusions, and simplified their model of deactivation.

Major Comments

#1 The new piece of information includes exploring the possibility of residue E130 driving pH sensitivity based on the structural data. This idea is quite interesting. Unfortunately, Fig 9A showed no current and the authors concluded that E130H "was not readily expressing in [our] oocytes." However, once co-expressed with KV7.3, they were able to record some currents. Although it is possible that KV7.3 "helps" with the expression of KV7.2 E130H mutant, another strong possibility is that KV7.2 E130H remains poorly expressed and/or trafficked to the cell membrane and that the current recorded is mostly from KV7.3 subunits only. This issue should be addressed before further conclusion can be drawn.

#2 Statistics is still missing in this version.

Minor Comments

#1 Although there are much fewer typos and grammatical errors in this revision compared to the original submission, there are still a significant number of typos and grammatical errors. Please proofread carefully prior to re-submission.

[...] although Reviewer #1 is satisfied with the revised manuscript, Reviewers #2 and #3 have raised a several issues about the data presented, and both again request that detailed statistical analyses of the data be presented throughout and that you carefully review the manuscript for clarity and readability. The editors concur with the reviewers' comments and recommendations.

We would be pleased to receive a suitably revised manuscript that addresses the concerns noted in the critiques and summarized above. While the editors agree with reviewer #1 and #2 that "the E130 mutations would have been more informative if they were made in the background of R198Q instead of the wild-type background", we don't require that these additional experiments be completed. Based on the scope of the requested changes, we would anticipate that the revision process will take no longer than 2 months.[...]

Thank you for the comments. We have addressed the reviewers' concerns to the best of our knowledge and abilities. Please see below.

In addition, we found the paragraph starting with the sentence "What is the role of R198 in open channel stabilization?" to be highly speculative. So, we decided to remove most of its content. Text in red font.

Reviewer #1 (Comments to the Authors):

The authors have responded well to my comments. However, the E130 mutations would have been more informative if they were made in the background of R198Q instead of wt background.

Reviewer #2 (Comments to the Authors):

The revised manuscript is much improved. For instance, the authors have extensively expanded the methods section to add previously missing information and have revised the discussion section and model to better align with their experimental observations of deactivation kinetics. However, some sections are still difficult to follow and some of my previous comments have not been fully addressed. Further adjustments would improve the quality of the work and make the conclusions easier to understand:

1. I find the authors response to my previous comment #3 (how the mutation induces hyperexcitability) and their associated discussion (row 461-482) difficult to follow. If I understand it correctly, the authors conclude that the disease phenotype of the R198Q mutant relates to the role of the Kv7.2/Kv7.3 channel in establishing the AHP, and that the slower activation kinetics of the mutant delay establishment of the AHP. I do not fully understand how they come to the conclusion that this is the anticipated functional effect of the mutant during neuronal firing given the mutant's complex behavior compared to WT (shifted V1/2, altered activation kinetics, dramatically altered deactivation kinetics). Additionally, how does this relate to the conclusion made on row 100-106, which rather points to the altered deactivation kinetics as the underlying cause?

We thank the reviewer for bringing this issue to our attention. The point made by the reviewer is well taken and we agree that this part of the discussion is confusing – at best – as it is written in the revised submission. To clarify the point we wanted to make, we speculate about the mechanism by which the enhanced activity of the mutant R198Q could result in hyperactivity. In rows 100-106, we proposed that an activity-induced decrease in pH_{EXT} would make the K^+ conductance less robust during repolarization when the mutant R198Q instead of the WT channel. In rows 461-482, we went further to acknowledge the expectation of an increased K^+ conductance during AHP, with $Kv7$ channels contributing to it. For the WT $Kv7.2$ channels, it is expected that the activity of $Kv7.2/Kv7.3$ would increase progressively during the lifetime of an AP burst as more of these channels will be activated due to the repetitive depolarizations. For the mutant R198Q, a similar situation would be expected. However, the contribution of this channel to the K^+ conductance would eventually decrease as pH_{EXT} decreases due to activity. In addition, the hyperpolarization caused by the enhanced activity of the mutant $Kv7.2$

channels, would promote a faster recovery of Nav channels. Consequently, the Na⁺ conductance would become more robust, so increasing excitability.

We accept that the writing in the paragraph in question is poor, so we thank again the reviewer for bringing this issue to our attention. The paragraph has been edited, showing the changes in blue font.

2. The authors responded to my previous comment #4 (missing statistics) that they had addressed this throughout the manuscript. However, I cannot find statistics supporting any of the authors claims. For instance, no statistics is given to support claims of shifts/no shifts in V1/2, or changes to kinetics. Several of the authors claims are convincing even without statistics, but other claims would be strengthened by inclusion of appropriate statistical calculations (such as T tests or ANOVA's). On row 526, the authors state that they use T tests to calculate statistics for time constraints. Where is the outcome of this statistical analysis reported?

We apologize for this omission. We are including the statistical tests on the figures in this version of the manuscript.

3. Good that the authors now include information about the Fermi-Dirac fit in the methods section. What is the reason for not including the outcome of the fit, in the form of lines, in the Itail-Vact plots? Without the fits, it is hard for the reader to judge how well the data is fitted.

We apologize for this omission too. There was no reason not to include plots of the fitted function, except for preventing the figures from being overloaded. We have included the fitted functions in this revised version.

4. For the structural model included in the revision (row 279-281), please include appropriate information in the methods section.

Thank you to the reviewer for pointing this out. We have included the requested information in the Methods section.

5. I liked the suggestion made by Reviewer 1 to test mutating residue E130 in a R198Q background, to see if this removes the pH effect. However, as far as I understand, the authors tested a E130H mutation in the WT background. What was the reason for this? Given that the WT channel shows limited pH effects, it is not clear to me what the authors expected from the E130H mutation alone. Also, because of the poor expression/function of the E130H mutant, an alternative would be to test the E130C or E130R mutations in a R198Q background, as the E130C and E130R mutants have been previously successfully studied using electrophysiology (Soldovieri et al., 2019, doi: 10.3390/ijms20143382).

In hindsight, we agree that the mutation E130H in the background could be more informative. However, we tried that mutation alone first to test the idea that the charge on residue 198 was important for gating, and that a counter-charged residue could serve to stabilize the closed state. As we show, that was not the case. As we are preparing this new version of the manuscript, we performed a few preliminary recordings using the double mutant and found it to be also pH_{EXT} sensitive. These preliminary results indicated no substantial changes in the effect of the mutation other than quantitative differences.

6. Row 461-462: "If the voltage-dependence for activation is similar for Kv7.2/Kv7.3 and for Kv7.2-R198Q/Kv7.3..." This sentence can be misinterpreted. The voltage dependence is clearly different for the WT and mutant heteromers at physiological pH (pH 7.4), as shown in Figure 7. However, $V_{1/2}$ of mutants approach that of WT during acidification. The sentence can be re-phrased to clarify this.

We agree with the reviewer and have edited the sentence accordingly. The sentence can be simplified to simply ask "How can the mutation KV7.2-R198Q lead to hyperactivity?". Text in red.

7. Figure 7 legend: The figure legend does not match the figure itself, which has panel A-H.

Thank you for pointing this out. We apologize for this distraction. The text of the legend has been updated.

8. There are many grammatical errors and unclear language throughout the text, which requires proofreading.

The manuscript has been proofread to the best of our abilities.

Reviewer #3 (Comments to the Authors):

In this revised manuscript, Mehrdel and Villalba-Galea have added a new experiment, rephrased the discussion conclusions, and simplified their model of deactivation.

Major Comments

#1 The new piece of information includes exploring the possibility of residue E130 driving pH sensitivity based on the structural data. This idea is quite interesting. Unfortunately, Fig 9A showed no current and the authors concluded that E130H "was not readily expressing in [our] oocytes." However, once co-expressed with KV7.3, they were able to record some currents. Although it is possible that KV7.3 "helps" with the expression of KV7.2 E130H mutant, another strong possibility is that KV7.2 E130H remains poorly expressed and/or trafficked to the cell membrane and that the current recorded is mostly from KV7.3 subunits only. This issue should be addressed before further conclusion can be drawn.

Thank you for the feedback from the reviewer. It is well-known in the community that expression of Kv7.3 channels alone results in small currents, much smaller than those from Kv7.2 expressed alone (e.g.: PMID: 9836639, 15483133). For this reason, given the magnitude of the currents observed and based on our own experience, we did not directly address this, as we considered to be a known issue. Nonetheless, we have included a statement in the manuscript to address the reviewer's concern (text in orange).

#2 Statistics is still missing in this version.

We have addressed this issue. We apologize for our distraction in the previous version.

Minor Comments

#1 Although there are much fewer typos and grammatical errors in this revision compared to the original submission, there are still a significant number of typos and grammatical errors. Please proofread carefully prior to re-submission.

The manuscript has been proofread to the best of our abilities.

December 1, 2023

Dr. Carlos A Villalba-Galea
University of the Pacific
Physiology and Pharmacology
3601 Pacific Ave
Stockton, CA 95211

Re: 202213284R2

Dear Carlos,

Thank you for submitting your revised manuscript, titled "Effect of a sensing charge mutation on the deactivation of KV7.2 channels" to JGP. Your manuscript has now been seen by three (3) reviewers, whose comments are appended below. As you will see, the reviewers are enthusiastic about the work and, for the most part, are satisfied with the revisions made in response to their previous comments and concerns. There is, however, one additional issue that needs your attention. Reviewer #2 has requested that you include a description of your statistical analysis in the methods section. The editors concur with this recommendation.

We look forward to receiving a revised manuscript that includes this additional information. Please submit your revised manuscript via the link below, along with a copy of the text with alterations highlighted (boldfaced or underlined).

Please pay particular attention to recent changes to our instructions to authors in the following sections: Data presentation, Blinding and randomization and Statistical analysis, under Materials and Methods, as shown here: <https://rupress.org/jgp/pages/submission-guidelines#prepare>. Also, To improve the reproducibility of published content, we have partnered with SciScore. Authors are prompted in eJP to copy and paste the Materials and Methods section of their manuscript for a SciScore assessment when submitting their revised manuscript. Authors are also encouraged to further revise their Materials and Methods if the SciScore is below 4. More information can be found here: <https://rupress.org/jgp/pages/submission-guidelines#sciscore>.

Please also note that JGP now requires authors to submit Source Data used to generate figures containing gels and Western blots with all revised manuscripts (when applicable). This Source Data consists of fully uncropped and unprocessed images for each gel/blot displayed in the main and supplemental figures. If your paper includes cropped gel and/or blot images, please be sure to provide one Source Data file for each figure that contains gels and/or blots along with your revised manuscript files. File names for Source Data figures should be alphanumeric without any spaces or special characters (i.e., SourceDataF#, where F# refers to the associated main figure number or SourceDataFS# for those associated with Supplementary figures). The lanes of the gels/blots should be labeled as they are in the associated figure, the place where cropping was applied should be marked (with a box), and molecular weight/size standards should be labeled wherever possible. Source Data files will be made available to reviewers during evaluation of revised manuscripts and, if your paper is eventually published in JGP, the files will be directly linked to specific figures in the published article.

Source Data Figures should be provided as individual PDF files (one file per figure). Authors should endeavor to retain a minimum resolution of 300 dpi or pixels per inch. Please review our instructions for export from Photoshop, Illustrator, and PowerPoint here: <https://rupress.org/jgp/pages/submission-guidelines#revised>

While you are revising your manuscript, we ask that you consider whether you have any artwork that might be suitable for the cover of JGP. Microscopy images are particularly good for cover artwork, but other types of image can be very effective, so we encourage you to be creative. Please don't restrict yourself to images from the paper; an image that is relevant to the work described would be just as suitable. Images should be a minimum resolution of 300 dpi. To see recent examples, visit the following page and click on 'Show covers? Yes': <https://jgp.rupress.org/content/by/year>

Thank you for submitting this paper to JGP.

Please submit your revised manuscript, and any associated files, via this link:
Link Not Available

Sincerely,

Jeanne

Jeanne Nerbonne, Ph.D.
On behalf of Journal of General Physiology

Journal of General Physiology's mission is to publish mechanistic and quantitative molecular and cellular physiology of the highest quality; to provide a best-in-class author experience; and to nurture future generations of independent researchers.

Reviewer #1 (Comments to the Authors):

The authors have responded well to my comments

Reviewer #2 (Comments to the Authors):

The authors have done a good job in revising the manuscript and responding to my comments. My only remaining recommendation is that the authors include a section describing their statistical analysis in the methods.

Reviewer #3 (Comments to the Authors):

The authors have adequately addressed the comments by the reviewer.

December 5, 2023

Dr. Carlos A Villalba-Galea
University of the Pacific
Physiology and Pharmacology
3601 Pacific Ave
Stockton, CA 95211

Re: 202213284R3

Dear Carlos,

I am pleased to let you know that your manuscript, titled "Effect of a sensing charge mutation on the deactivation of KV7.2 channels" is scientifically acceptable for publication in Journal of General Physiology. Formal acceptance will follow when it is modified in accordance with our editorial policies.

Please note items that need attention are listed at the bottom of this email (under 'manuscript formatting checklist'). Please also be sure to include a copy of the text with alterations highlighted (boldfaced or underlined). Your manuscript should be a double-spaced MS Word file and include editable tables, if appropriate.

JGP now requires a data availability statement for all research article submissions. These statements will be published in the article directly above the Acknowledgments. The statement should address all data underlying the research presented in the manuscript. Please visit the JGP instructions for authors for guidelines and examples of statements at <https://rupress.org/jgp/pages/editorial-policies#data-availability-statement>.

Lastly, JGP adds short captions to articles listed on our weekly newest article emails. If you haven't, please provide a short, ~40-word summary statement for the online JGP table of contents and alerts. This summary should describe the context and significance of the findings for a general readership and be placed on/near the title page.

Please submit your final files via this link:
Link Not Available

Thank you for choosing to publish your research in JGP and please feel free to contact me with any questions.

Sincerely,

Jeanne

Jeanne Nerbonne, Ph.D.
On behalf of Journal of General Physiology

Journal of General Physiology's mission is to publish mechanistic and quantitative molecular and cellular physiology of the highest quality; to provide a best in class author experience; and to nurture future generations of independent researchers.

Manuscript formatting checklist:

- MS Word document of text needed (including editable tables)
- MS Word document of supplemental text needed, if applicable (including figure legends and editable tables)
- Brief Statement describing supplementary information needed, if applicable (in subsection at end of Materials & Methods)
- Please include a data availability statement preceding the Acknowledgments section. Please see <https://rupress.org/jgp/pages/editorial-policies#data-availability-statement>
- Figures created at sufficient resolution and in acceptable format (including supplemental if applicable). If working in Illustrator, we prefer .ai or .eps file format. If working in Photoshop please use 600dpi/1000dpi .tiff or .psd file format. Minimum resolution at estimated print size: Minimum resolution for all figures is 600 dpi. For figures that contain both photographs and line art or text, 600 dpi is highly recommended. Figures containing only black and white elements (line art, no color, and no gray) should be 1,000 dpi. Maximum figure size is 7 in wide x 9 in high (17.5 x 22.8 cm) at the correct resolution. <https://jgp.rupress.org/fig-vid-guidelines>
- Supplemental figures, if any, conforming to same guidelines as manuscript figures (noted above)
- If images resemble one from a prior publications, the author must seek permissions (to reproduce or adapt) from the original publisher. [You can resubmit your paper while waiting to hear back from the original publisher but please keep us updated]

- All authors must complete a disclosure form prior to acceptance. A link to complete the form has been sent to all coauthors. Please provide the editorial office with updated email addresses if necessary